# ProteinGym: Large-Scale Benchmarks for Protein Fitness Prediction and Design

**Pascal Notin**[†][*]
Computer Science,
University of Oxford

**Aaron W. Kollasch**[†]
Systems Biology,
Harvard Medical School

**Daniel Ritter**[†]
Systems Biology,
Harvard Medical School

**Lood van Niekerk**[†]
Systems Biology,
Harvard Medical School

**Steffanie Paul**
Systems Biology,
Harvard Medical School

**Hansen Spinner**
Systems Biology,
Harvard Medical School

**Nathan Rollins**
Seismic Therapeutic

**Ada Shaw**
Applied Mathematics,
Harvard University

**Ruben Weitzman**
Computer Science,
University of Oxford

**Jonathan Frazer**
Centre for Genomic Regulation
Universitat Pompeu Fabra

**Mafalda Dias**
Centre for Genomic Regulation
Universitat Pompeu Fabra

**Dinko Franceschi**
Systems Biology,
Harvard Medical School

**Rose Orenbuch**
Systems Biology,
Harvard Medical School

**Yarin Gal**
Computer Science,
University of Oxford

**Debora S. Marks**[*]
Harvard Medical School
Broad Institute

## Abstract

Predicting the effects of mutations in proteins is critical to many applications, from understanding genetic disease to designing novel proteins that can address our most pressing challenges in climate, agriculture and healthcare. Despite a surge in machine learning-based protein models to tackle these questions, an assessment of their respective benefits is challenging due to the use of distinct, often contrived, experimental datasets, and the variable performance of models across different protein families. Addressing these challenges requires scale. To that end we introduce ProteinGym, a large-scale and holistic set of benchmarks specifically designed for protein fitness prediction and design. It encompasses both a broad collection of over 250 standardized deep mutational scanning assays, spanning millions of mutated sequences, as well as curated clinical datasets providing high-quality expert annotations about mutation effects. We devise a robust evaluation framework that combines metrics for both fitness prediction and design, factors in known limitations of the underlying experimental methods, and covers both zero-shot and supervised settings. We report the performance of a diverse set of over 70 high-performing models from various subfields (eg., alignment-based, inverse folding) into a unified benchmark suite. We open source the corresponding codebase, datasets, MSAs, structures, model predictions and develop a user-friendly website that facilitates data access and analysis.

[*]Correspondence: pascal.notin@cs.ox.ac.uk, awkollasch@gmail.com, danieldritter1@gmail.com, loodvn@gmail.com, debbie@hms.harvard.edu ; † Equal contribution

37th Conference on Neural Information Processing Systems (NeurIPS 2023) Track on Datasets and Benchmarks.

# 1 Introduction

Proteins carry out a wide range of functions in nature, facilitating chemical reactions, transporting molecules, signaling between cells, and providing structural support to cells and organisms. This astonishing functional diversity is uniquely encoded in their amino acid sequence. For instance, the number of possible arrangements for a 64-residue peptide chain ($20^{64}$) is already larger than the estimated number of atoms in the universe. Despite substantial progress in sequencing over the past two decades, we have observed a relatively small, biased portion of that massive sequence space. Consequently, the ability to manipulate and optimize known sequences and structures represents tremendous opportunities to address pressing issues in climate, agriculture and healthcare.

The design of novel, functionally optimized proteins presents several challenges. It begins with learning a mapping between protein sequences or structures and their resulting properties. This mapping is often conceptualized as a "fitness landscape", a multivariate function that characterizes the relationship between genetic variants and their adaptive fitness. The more accurately and comprehensively we can define these landscapes, the better our chances of predicting the effects of mutations and designing proteins with desirable and diverse properties. Machine learning, by modeling complex, high-dimensional relationships, has emerged as a powerful tool for learning these fitness landscapes. In recent years, a plethora of machine learning methods have been proposed for protein modeling, each promising to offer new insights into protein function and design. However, assessing the effectiveness of these methods has proven challenging. A key issue is their evaluation on distinct and relatively sparse benchmark datasets, while relative model performance fluctuates importantly across experimental assays, as was shown in several prior analyses [Riesselman et al., 2018, Laine et al., 2019, Meier et al., 2021]. This situation underscores the importance of scale in the benchmarks used. Larger, more diverse datasets would offer a more robust and comprehensive evaluation of model performance.

To address these limitations, we introduce ProteinGym, a large-scale set of benchmarks specifically tailored to protein design and fitness prediction. It comprises a broad collection of over 250 standardized Deep Mutational Scanning (DMS) assays which include over 2.7 million mutated sequences across more than 200 protein families, spanning different functions, taxa and depth of homologous sequences. It also encompasses clinical benchmarks providing high-quality annotations from domain experts about the effects of ∼65k substitution and indel mutations in human genes (§ 3).

We have designed ProteinGym to be an effective, holistic, robust, and user-friendly tool. It provides a structured evaluation framework that factors in known limitations of the underlying experimental methods and includes metrics that are tailored to protein design and mutation effect prediction (§ 4). We report the performance in a unified benchmark of over 70 diverse high-performing models that come from various subfields of computational biology (eg., mutation effects prediction, sequence-based models for de novo design, inverse folding), thereby supporting novel comparisons across. Unlike prior benchmarks, ProteinGym integrates both the zero-shot and supervised settings, leading to new insights (§ 5). All models are codified with a common interface in the same open-source codebase, promoting consistency and ease of use. Lastly, a dedicated website offers an interactive platform to facilitate comparisons across datasets and performance settings.

# 2 Related Work and Background

**Multi-task protein benchmarks**   In recent years, several benchmarks have been introduced to provide initial means to assess protein model performance across a multitude of tasks of interests, e.g., predicting contacts, structure, thermostability, and fitness. These benchmarks are generally geared towards assessing the quality of learned protein representations, and the extent to which these representations can be broadly leveraged for various tasks. However, for fitness prediction, they all rely on a very limited set of proteins (e.g., 1-3 assays). In comparison, the ProteinGym benchmarks focus on a single task – fitness prediction – and encompass *two orders of magnitude more* point mutations assessed and vast diversity of protein families included.

TAPE (Tasks Assessing Protein Embeddings) [Rao et al., 2019] covers five protein prediction tasks, each designed to test a different aspect of protein function and structure prediction (secondary structure, contact, remote homology, fluorescence and stability), and focuses on assessments in the semi-supervised regime via carefully curated train-validation-test splits. ProteinGLUE [Capel et al.,

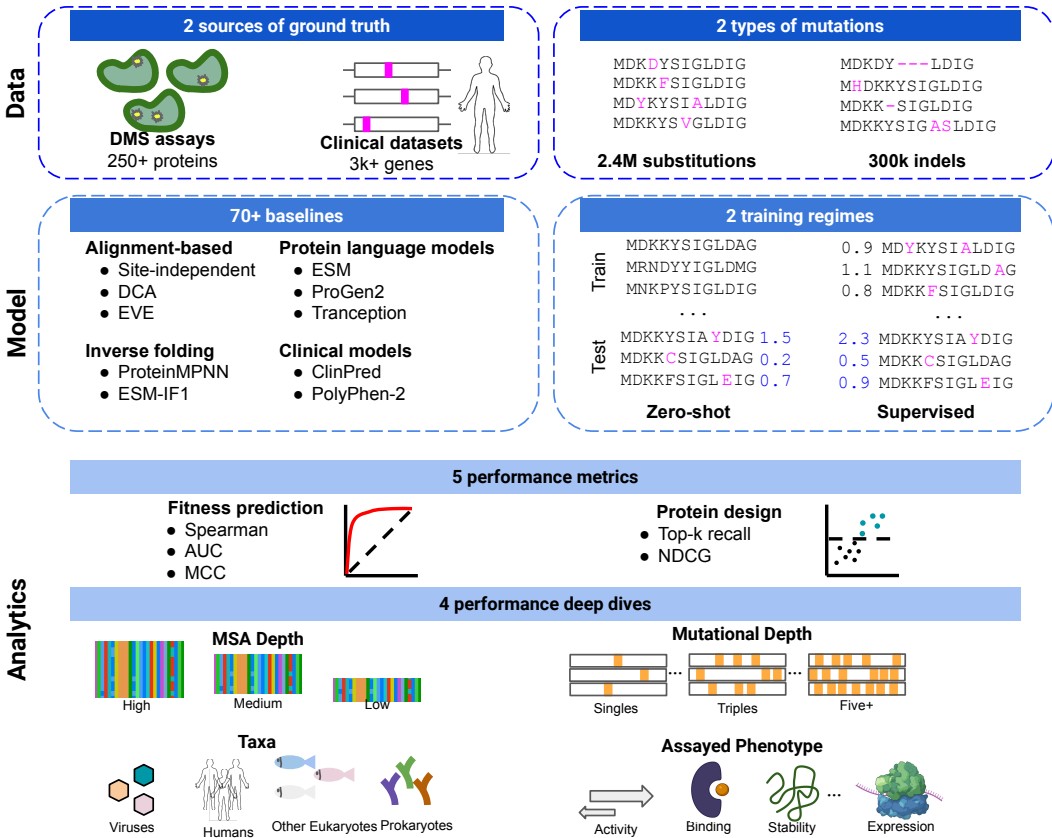

Figure 1: **ProteinGym benchmarks** ProteinGym is comprised of three layers. The data layer encompasses two complementary ground truth labels from DMS assays and clinical annotations from experts. For both, we analyze two types of mutations: substitutions and indels. The model layer is comprised of a diverse set of baselines, tailored to both zero-shot and supervised training regimes. Lastly, the analytics layer includes several performance metrics geared towards fitness prediction or protein design evaluation. Different segmentation variables (e.g., MSA depth, assayed phenotype, taxa) facilitate the comparisons of models across diverse settings

2022] also focuses on assessing the usefulness of learned protein representations on supervised downstream tasks. It is comprised of five different tasks, none directly related to protein fitness: secondary structure, solvent accessibility, protein-protein interactions, epitope region and hydrophobic patch prediction. PEER [Xu et al., 2022] also focuses on multi-task benchmarking, grouped in five categories: protein property, localization, structure, protein-protein interactions and protein-ligand interactions. It contains a richer set of evaluations compared with the prior two benchmarks, and also investigates the multi-task learning setting, but is not designed for thorough fitness prediction benchmarking (3 fitness related assays). The handful of fitness DMS assays from these various benchmarks are all subsumed in ProteinGym.

**Single task, non-fitness datasets & benchmarks** Efforts to create fair, large-scale, and comprehensive benchmarks have been a significant focus of computational biologists for certain tasks. Among these, the biennial Critical Assessment of protein Structure Prediction (CASP) [Kryshtafovych et al., 2021] is the most renowned. CASP concentrates on protein structure prediction and has set the gold standard in this domain. In parallel to CASP, the Critical Assessment of Functional Annotation (CAFA) [Zhou et al., 2019] challenge provides a platform for evaluating protein function classification. The SKEMPI [Moal and Fernández-Recio, 2012] database is specifically designed to aid the evaluation of computational methods predicting the effect of mutations on protein-protein binding affinity. Several datasets have been curated for specific properties of interest across a diverse set of

proteins, for instance thermostability [Tsuboyama et al., 2023, Stourac et al., 2020, Chen et al., 2022] or solubility [Hon et al., 2020].

**Protein fitness benchmarks**   Closest to our work are the collections of DMS assays that were curated in Hopf et al. [2017] (28 substitution assays), and then further expanded upon in Riesselman et al. [2018] (42 substitution assays) and Shin et al. [2021] (4 indel assays). We include all assays related to fitness prediction from these prior works in ProteinGym. FLIP [Dallago et al., 2021] focused on comparing fitness predictors in the semi-supervised setting, developing a robust evaluation framework and curating cross-validation schemes for three assays. MaveDB [Rubin et al., 2021] is a repository rather than a benchmark, but it compiles a large collection of datasets from multiple variant effect mapping experiments that can be used for benchmarking purposes. An initial prototype of the ProteinGym benchmarks (referred to as 'ProteinGym v0.1') was introduced in Notin et al. [2022a]. We have since then significantly expanded the benchmarks in terms of number and diversity of underlying datasets, baselines, evaluation framework and model training regimes (Table A1). This not only enables performance evaluation at an unprecedented scale, but also builds connections between different subfields that are often perceived as separate, as we discuss in the following paragraph.

**Clinical Benchmarks**   Designing an unbiased, non-circular and broadly applicable benchmark to evaluate the performance of human variant effect predictors at predicting clinical significance is still an open-problem for the clinical community. Combining DMS with clinical annotations has been a fruitful direction to avoid biases [Frazer et al., 2021, Livesey and Marsh, 2023]. ClinGen curated a clinical dataset specifically designed to compare a subset of models [Pejaver et al., 2022].

**Relationship between protein fitness, mutation effect prediction and design**   The protein fitness landscape refers to the mapping between genotype (e.g., the amino acid sequence) and phenotype (e.g., protein function). While it is a fairly broad concept, it should always be thought about in practice within a particular context (e.g., stability at a given temperature in a specific organism). Models that learn the protein fitness landscape have been shown to be effective at predicting the effects of mutations [Frazer et al., 2021, Jagota et al., 2022, Brandes et al., 2023, Notin et al., 2022b]. But the ability to tell apart the sequences that are functional or not is also critical to protein engineering efforts [Romero et al., 2012, Yang et al., 2018, Wu et al., 2019, Alley et al., 2019b]. Although typically introduced in the context of de novo protein design [Huang et al., 2016], inverse folding methods [Ingraham et al., 2019, Jing et al., 2020, Dauparas et al., 2022, Gao et al., 2022] can also be used for mutation effects prediction (Appendix A.4.1). There is thus a very tight connection between protein fitness, mutation effect prediction and protein engineering, and the same models can be used for either task depending on context. We seek to illustrate this connection through this work, comparing baselines introduced in different fields (e.g., protein representation learning, inverse folding models, co-evolution models) on the same benchmarks, and including different metrics that are geared more to mutation effect prediction (e.g., Spearman) or design tasks (e.g., NDCG).

## 3   ProteinGym benchmarks

ProteinGym is a collection of benchmarks (Fig. 1) that cover different types of mutation (ie., substitutions vs. indels), ground-truth labels (ie., experimental measurement from DMS vs. clinical annotations), and model training regime (ie., zero-shot vs. supervised).

### 3.1   Mutation types

We curate benchmarks for two types of protein mutations – *substitutions* and *indels* (insertions or deletions), each with unique implications for the structure, function, and modeling of proteins.

**Substitutions**   Substitution mutations refer to a change in which one amino acid in a protein sequence is replaced by another. Depending on the properties of the substituted amino acid, this can have varied impacts on the protein's structure and function, which can range from minimal to drastic. The influence of a substitution largely depends on whether it is conservative (i.e., the new amino acid shares similar properties to the original) or non-conservative. In terms of computational modeling, substitutions are the most commonly addressed mutation type, and the majority of mutation effect predictors support substitutions.

**Indels**   Indel mutations correspond to insertions or deletions of amino acids in protein sequences. While indels can affect protein fitness in similar ways to substitutions, they can also have profound impacts on protein structure by altering the protein backbone, causing structural modifications inaccessible through substitutions alone [Shortle and Sondek, 1995, Tóth-Petróczy and Tawfik, 2013]. From a computational perspective, indels present a unique challenge because they alter the length of the protein sequence, requiring additional considerations in model design and making it more difficult to align sequences. For instance, the majority of models trained on Multiple Sequence Alignments are typically unable to score indels due to the fixed coordinate system they operate within (see § 4). Furthermore, when dealing with probabilistic models, comparing relative likelihoods of sequences with different lengths results in additional complexities and considerations.

## 3.2   Dataset types

The fitness of a protein is a measure of how well a protein can perform its function within an organism. Factors that influence protein fitness are diverse and include stability, folding efficiency, catalytic activity (for enzymes), binding specificity and affinity. To properly capture this diversity, we curated a broad set of experimental assays that map a given sequence to phenotypic measurements that are known or hypothesized to be related to its fitness. We focused on two potential sources of ground truth: Deep Mutational Scanning (DMS) assays and Clinical datasets.

**Deep Mutational Scanning assays**   Modeling protein fitness landscapes presents a challenge due to the complex relationship between experimentally measured protein fitness, the distribution of natural sequences, and the underlying fitness landscape. It is challenging to isolate a singular, measurable molecular property that reflects the key aspects of fitness for a given protein. In developing ProteinGym, we prioritized assays where the experimentally measured property for each mutant protein is likely to represent the role of the protein in organismal fitness. The resulting compilation of over 250 DMS assays extends over a wide array of functional properties, including ligand binding, aggregation, thermostability, viral replication, and drug resistance. It encompasses diverse protein families, such as kinases, ion channel proteins, G-protein coupled receptors, polymerases, transcription factors, and tumor suppressors. In contrast to most DMS assay collections that focus exclusively on single amino acid substitutions, ProteinGym includes several assays with multiple amino acid variants. Moreover, it spans different taxa (i.e., humans, other eukaryotes, prokaryotes, and viruses), alignment depths, and mutation types (substitutions vs indels). All details about the curation and pre-processing of these DMS assays are provided in Appendix A.3.

**Clinical datasets**   ClinVar [Landrum and Kattman, 2018] is an extensive, public database developed by the National Center for Biotechnology Information (NCBI). It serves as an archival repository that collects and annotates reports detailing the relationships among human genetic variations and associated phenotypes with relevant supporting evidence, thereby providing robust, clinically annotated datasets that are invaluable for understanding the functional impact of mutations. From the standpoint of benchmarking mutation effects predictors, ClinVar permits the direct comparison of predictive models in terms of their accuracy in estimating the functional impact of mutations on human health. Annotations are also available for an order of magnitude more distinct proteins compared with our DMS-based benchmarks, albeit much sparser per protein (see table 1). In the case of indels, we focused on short ($\leq 3$ amino acids) variants. In ClinVar, 84% of indel annotations are pathogenic, so we added to our clinical dataset common indels from gnomAD (allele frequency >5%) as pseudocontrols [Karczewski et al., 2020].

## 3.3   Model training regime

Lastly, we discriminate in our benchmarks between zero-shot and supervised settings. In the supervised regime we are allowed to leverage a subset of labels to train a predictive model, while in the zero-shot setting we seek to predict the effects of mutations on fitness without relying on the ground-truth labels for the protein of interest. These two settings offer complementary viewpoints of practical importance. For instance, in settings where labels are subject to several biases or scarcely available (e.g., labels for rare genetic pathologies), we need methods with robust zero-shot performance performance. In cases where we seek to design new proteins that simultaneously optimize several properties of interest (e.g., binding affinity, thermostability) and we have collected a sufficiently large number of labels for each target, supervised methods are more appropriate. The

| Dataset | Description | Mutation type | # Proteins | # Mutants |
|---------|-------------|---------------|------------|-----------|
| DMS | High-throughput assays evaluating the functional impact of a wide range of protein mutations | Substitutions
Indels | 217
66 | 2.4M
0.3M |
| Clinical | Expert-curated clinical annotations across a wide range of human genes | Substitutions
Indels | 2,525
1,555 | 63k
3k |
| Total | | | 3,422 | 2.7M |

Table 1: **ProteinGym datasets summary** ProteinGym includes a large collection of DMS assays and clinical datasets that offer complementary viewpoints when assessing protein fitness. The table reports the number of mutants and unique proteins per dataset (the total being deduped across datasets).

need to rely on labels is even more pronounced when we seek to optimize several anti-correlated properties or when evolution is a poor proxy for the property of interest. Predictions obtained in the zero-shot settings may also be used to augment supervised models [Hsu et al., 2022a]. The two settings require substantially different evaluation frameworks, which we detail in § 4.

# 4 Evaluation framework

## 4.1 Zero-shot benchmarks

**DMS assays**    In the zero-shot setting we predict experimental phenotypical measurements from a given assay, without having access to the labels at training time. Due to the often non-linear relationship between protein function and organism fitness [Boucher et al., 2016], the Spearman's rank correlation coefficient is the most generally appropriate metric for model performance on experimental measurements. We use this metric similarly to previous studies [Hopf et al., 2017, Riesselman et al., 2018, Meier et al., 2021]. However, in situations where DMS measurements exhibit a bimodal profile, rank correlations may not be the optimal choice. Consequently, for these instances, we supplement our performance assessment with additional metrics, namely the Area Under the ROC Curve (AUC), and the Matthews Correlation Coefficient (MCC), which compare model scores with binarized experimental measurements. Furthermore, for certain goals (e.g., optimizing functional properties of designed proteins), it is more important that a model is able to correctly identify the most functional protein variants, rather than properly capture the overall distribution of all assayed variants. Thus, we also calculate the Normalized Discounted Cumulative Gains (NDCG), which up-weights a model if it gives its highest scores to sequences with the highest DMS value. We also calculate Top K Recall, where we select K to be the top 10% of DMS values. To avoid placing too much weight on properties where we have many assays (e.g., thermostability), we first compute each of these metrics within groups of assays that measure similar functions. The final value of the metric is then the average of these averages, giving each functional group equal weight. We refer to the corresponding value as 'corrected average'.

**Clinical datasets**    For the clinical data, with pathogenic and benign categories, we calculate the areas under the ROC and precision-recall curves. In the substitution dataset, 50% of the labels are in approximately 10% of the proteins. Since clinical labels across genes correspond to underlying pathologies that are very distinct to one another, it is preferable to assess performance on a gene-by-gene basis. We thus compute the average per-gene performance on the substitution benchmark. However, in the case of indels, only about half of the proteins has a pathogenic label (and only 10% have a both pathogenic and benign or pseudocontrol labels), so we compute the total AUC for the full dataset. The problem of calibrating model scores in a principled way across different genes is an open problem; we leave this to future work.

**Baselines**    We implement a diverse set of 50+ zero-shot baselines that may be grouped into alignment-based models, protein language models, inverse folding models and 'hybrid' models. Alignment-based models, such as site-independent and EVmutation models [Hopf et al., 2017], DeepSequence [Riesselman et al., 2018], WaveNet [Shin et al., 2021], EVE [Frazer et al., 2021] and GEMME [Laine et al., 2019], are trained on Multiple Sequence Alignments (MSAs). Protein language models are trained on large quantities of unaligned sequences across protein families. They

include UniRep [Alley et al., 2019a], the RITA suite [Hesslow et al., 2022], the ESM1 and ESM2 suite [Rives et al., 2021, Meier et al., 2021, Lin et al., 2023], VESPA [Marquet et al., 2022], the CARP suite [Yang et al., 2023a] and the ProGen2 suite [Nijkamp et al., 2022]. Inverse Folding models learn sequence distributions conditional on an input structure [Ingraham et al., 2019]. We include here ProteinMPNN [Dauparas et al., 2022] which is trained on structures in the PDB, MIF [Yang et al., 2023b] trained on CATH4.2 [Dawson et al., 2016], and ESM-IF1 [Hsu et al., 2022b] which is trained on the PDB and a dataset of AlphaFold2 folded structures. Hybrid models combine the respective strengths of family-specific alignment-based and family-agnostic language models, such as the MSA Transformer [Rao et al., 2021], evotuned UniRep [Alley et al., 2019a], Tranception [Notin et al., 2022a] and TranceptEVE [Notin et al., 2022b].

Because of the variable length of sequences subject to insertion or deletion mutations, alignment-based methods with fixed matrix representations of sequences are unable to score indels. However, profile Hidden Markov Model (HMM) and autoregressive models include explicit or implicit probabilities of indels at each position. Both are trained on homologous sequences recovered with an MSA and expanded to include insertions. The masked-marginals heuristic Meier et al. [2021] used to predict protein fitness with protein language models trained with a masked-language modeling objective (e.g., ESM-1v, MSA Transformer) does not support indels (see Appendix A.4). We thus only report the performance of the following baselines: Tranception [Notin et al., 2022a], TranceptEVE [Notin et al., 2022b], WaveNet [Shin et al., 2021], HMM [Eddy, 2011], ProGen2 [Madani et al., 2020], UniRep [Alley et al., 2019a], RITA [Hesslow et al., 2022] and ProtGPT2 [Ferruz et al., 2022].

For comparisons on clinical benchmarks, we also include unsupervised baselines developed for variant effect prediction in humans, such as SIFT [Ng and Henikoff, 2002], MutPred [Li et al., 2009], LRT [Chun and Fay, 2009], MutationAssessor [Reva et al., 2011], PROVEAN [Choi et al., 2012], PrimateAI [Sundaram et al., 2018] and LIST-S2 [Malhis et al., 2020].

## 4.2 Supervised benchmarks

**DMS assays** We leverage the same set of 250+ substitutions and indels DMS assays as for the zero-shot setting. In the supervised setting, greater care should be dedicated to mitigating overfitting risks, as the observations in biological datasets may not be fully independent. For instance, two mutations involving amino acids with similar biochemical properties at the same position will tend to produce similar effects. If we train on one of these mutations and test on the other, we will tend to overestimate our ability to predict the effects of mutants at unseen positions. In order to quantify the ability of each model to extrapolate to unseen positions at training time, we leverage 3 types of cross-validation schemes introduced in Notin et al. [2023]. In the *Random* scheme, each mutation is randomly assigned to one of five different folds. In the *Contiguous* scheme, we split the sequence contiguously along its length, in order to obtain 5 segments of contiguous positions, and assign mutations to each segment based on the position at which it occurs. Lastly, in the *Modulo* scheme, we assign positions to each fold using the modulo operator to obtain 5 folds overall. In all supervised settings, we report both the Spearman's rank correlation and Mean Squared Error (MSE) between predictions and experimental measurements. A more challenging generalization task would involve learning the relationship between protein representation (sequence, structure, or both) and function using only a handful of proteins, and then extrapolating at inference time to protein families not encountered during training. This setting may be seen as a hybrid between the zero-shot and supervised regimes – closer to zero-shot if we seek to predict different properties across families, and closer to the supervised setting if the properties are similar (eg., predicting the thermostability of proteins with low sequence similarity with the ones in the training set). While this study does not delve into these hybrid scenarios, the DMS assays in ProteinGym can facilitate such analyses.

**Clinical datasets** Given the restrictions on the number of labels available per gene and the discrepancies between train-validation-test splits across the different supervised baselines, we report test performance on the full set of all available ClinVar labels. We note that this may result in overestimating the performance of supervised methods for which the training data would substantially overlap with the labels considered in our ClinVar set. Further data leakage occurs for models trained on population frequencies, as most ClinVar benign labels are established based on observed frequencies in humans (situation especially evident for our indel dataset where we use frequent variants as pseudocontrols). Interestingly, despite this overfitting risk and as first observed in Frazer et al. [2021], we find that most supervised methods are outperformed by the best unsupervised methods (Fig. 2).

**Baselines** For the supervised DMS benchmark, we report two suites of baselines. The first suite is comprised of models that take as inputs One-Hot-Encoded (OHE) features. Following the protocol described in Hsu et al. [2022a], we augment the model inputs with predictions from several state-of-the-art zero-shot baselines: DeepSequence [Riesselman et al., 2018], ESM-1v [Meier et al., 2021], MSA Transformer [Rao et al., 2019], Tranception [Notin et al., 2022a] and TranceptEVE [Notin et al., 2022b]. Following prior works from the semi-supervised protein modeling literature [Heinzinger et al., 2019, Dallago et al., 2021], the second suite is formed with baselines that leverage mean-pooled embeddings from several protein language models (ESM-1v, MSA Transformer and Tranception) in lieu of OHE features. We also augment these baselines with zero-shot predictions obtained with the same model used to extract the protein sequence embeddings. Lastly, we include ProteinNPT [Notin et al., 2023], a semi-supervised pseudo-generative architecture which jointly models sequences and labels by performing axial attention [Ho et al., 2019b, Kossen et al., 2022] on input labeled batches. Additional details for the corresponding model architectures are reported in Appendix A.4.2. On the various clinical benchmarks, the above baselines are challenging to train given the low number of labels available per gene. We instead include several supervised baselines that have been specifically developed for variant effects predictions in humans, such as ClinPred [Alirezaie et al., 2018], MetaRNN [Li et al., 2022], BayesDel [Feng, 2017], REVEL [Ioannidis et al., 2016] and PolyPhen-2 [Adzhubei et al., 2010] (full list in A.4.3).

## 5 Results

### 5.1 Substitution benchmarks

We follow the experimental protocol described in § 4.1 and report our main results on the zero-shot DMS benchmarks in Table 2, supervised DMS benchmark in Table 3, and combined supervised and unsupervised clinical benchmarks in Fig. 2A. TranceptEVE emerges as the best overall method across the various settings. One of the key objectives of ProteinGym benchmarks is to analyze performance across a wide range of regimes to guide model selection depending on the objectives of the practitioners. To that end we also provide a performance breakdown across MSA depth, mutational depth and taxa where relevant (see Appendix A.5 and supplements). While TranceptEVE tops the ranking across the majority of metrics and settings, GEMME achieves the best performance in several categories, such as assays of viral or non-human eukaryotic proteins, and low and medium depth MSAs. While we report average performance per metric in Table 2, the *distribution* of scores across assays is also insightful. For instance, certain models are heavily penalized in aggregate rankings due to very poor performance on a handful of assays (e.g., ESM-1v), such that looking a the median performance in lieu of the average provides a complementary viewpoint. Furthermore, although most models rank similarly under Spearman and NDCG, some have comparatively better performance in one over the other (Fig. 2B). Superior ranking under NDCG may suggest a model is better at predicting the top end of a score distribution, which may be a desirable feature when using models for design and optimization. Many of the alignment-based methods (e.g. EVmutation, WaveNet) exhibit this behavior (Fig. A1). Models with higher relative Spearman (e.g., ESM-1v and ESM-2) may be more effective for cases where the model needs to learn the full property distribution well, such as with mutation effect prediction. Lastly, in the zero-shot setting, autoregressive protein language models (e.g., Tranception, ProGen2) tend to outperform their masked language modeling counterparts (e.g., ESM models). However, in supervised settings, both types of models provide valuable embeddings for learning. The optimal method depends on the specific situation, as observed in Table 3 and Table A16. The best performance is achieved with the ProteinNPT architecture, demonstrating the value from performing self-attention alternatively across columns (i.e., amino acid tokens and labels) and rows (i.e., protein sequences) to learn a rich representation of the data.

### 5.2 Indel benchmarks

The zero-shot results for an indel-compatible subset of the models in ProteinGym is shown in Table 4. The Spearman rank correlations are separated by the method used to generate test sequences: unbiased libraries, or model-designed sequences biased towards natural sequences. Model performance exhibits higher variance across assay types, with ProGen2 achieving the highest performance on Library assays (albeit with low performance on designed assays), WaveNet topping the ranking on designed assays (but with low performance on library assays), and TranceptEVE reaching high performance across both. We provide additional indel results in the supervised and clinical settings in Appendix A.5.

| Model type | Model name | Spearman | AUC | MCC | NDCG | Recall |
|---|---|---|---|---|---|---|
| Alignment-based | Site-Independent | 0.359 | 0.696 | 0.286 | 0.747 | 0.201 |
| | WaveNet | 0.373 | 0.707 | 0.294 | 0.761 | 0.203 |
| | EVmutation | 0.395 | 0.716 | 0.305 | 0.777 | 0.222 |
| | DeepSequence (ensemble) | 0.419 | 0.729 | 0.328 | 0.776 | 0.226 |
| | EVE (ensemble) | 0.439 | 0.741 | 0.342 | 0.783 | **0.230** |
| | GEMME | **0.455** | 0.749 | 0.352 | 0.777 | 0.211 |
| Protein language | UniRep | 0.190 | 0.605 | 0.147 | 0.647 | 0.139 |
| | CARP (640M) | 0.368 | 0.701 | 0.285 | 0.748 | 0.208 |
| | ESM-1b | 0.394 | 0.719 | 0.311 | 0.747 | 0.203 |
| | ESM-2 (15B) | 0.401 | 0.720 | 0.314 | 0.759 | 0.208 |
| | RITA XL | 0.372 | 0.707 | 0.293 | 0.751 | 0.193 |
| | ESM-1v (ensemble) | 0.407 | 0.723 | 0.320 | 0.749 | 0.211 |
| | ProGen2 XL | 0.391 | 0.717 | 0.306 | 0.767 | 0.199 |
| | VESPA | 0.436 | 0.742 | 0.346 | 0.775 | 0.201 |
| Hybrid | UniRep evotuned | 0.347 | 0.693 | 0.274 | 0.739 | 0.181 |
| | MSA Transformer (ensemble) | 0.434 | 0.738 | 0.340 | 0.779 | 0.224 |
| | Tranception L | 0.434 | 0.739 | 0.341 | 0.779 | 0.220 |
| | TranceptEVE L | **0.456** | **0.751** | **0.356** | **0.786** | **0.230** |
| Inverse Folding | ESM-IF1 | 0.422 | 0.730 | 0.331 | 0.748 | 0.223 |
| | MIF-ST | 0.401 | 0.718 | 0.311 | 0.766 | 0.226 |
| | ProteinMPNN | 0.258 | 0.639 | 0.196 | 0.713 | 0.186 |

Table 2: **Zero-shot substitution DMS benchmark** Corrected average of Spearman's rank correlation, AUC, MCC, NDCG@10%, and top 10% recall between model scores and experimental measurements on the ProteinGym substitution benchmark.

| Model type | Model name | Spearman ($\uparrow$) | | | | MSE ($\downarrow$) | | | |
|---|---|---|---|---|---|---|---|---|---|
| | | Contig. | Mod. | Rand. | Avg. | Contig. | Mod. | Rand. | Avg. |
| OHE | None | 0.064 | 0.027 | 0.579 | 0.224 | 1.158 | 1.125 | 0.898 | 1.061 |
| | DeepSequence | 0.400 | 0.400 | 0.521 | 0.440 | 0.967 | 0.940 | 0.767 | 0.891 |
| | ESM-1v | 0.367 | 0.368 | 0.514 | 0.417 | 0.977 | 0.949 | 0.764 | 0.897 |
| | MSAT | 0.410 | 0.412 | 0.536 | 0.453 | 0.963 | 0.934 | 0.749 | 0.882 |
| | Tranception | 0.419 | 0.419 | 0.535 | 0.458 | 0.985 | 0.934 | 0.766 | 0.895 |
| | TranceptEVE | 0.441 | 0.440 | 0.550 | 0.477 | 0.953 | 0.914 | 0.743 | 0.870 |
| Embed. | ESM-1v | 0.481 | 0.506 | 0.639 | 0.542 | 0.937 | 0.861 | 0.563 | 0.787 |
| | MSAT | 0.525 | 0.538 | 0.642 | 0.568 | 0.836 | 0.795 | 0.573 | 0.735 |
| | Tranception | 0.490 | 0.526 | 0.696 | 0.571 | 0.972 | 0.833 | 0.503 | 0.769 |
| NPT | ProteinNPT | **0.547** | **0.564** | **0.730** | **0.613** | **0.820** | **0.771** | **0.459** | **0.683** |

Table 3: **Supervised substitution DMS benchmark**. Corrected average of Spearman's rank correlation and MSE between model predictions and experimental measurements. MSAT is a shorthand for MSA Transformer.

## 6 Resources

**Codebase** A key contribution of our work is the consolidation of the numerous baselines discussed in § 4 in a single open-source GitHub repository. While the main code for the majority of these baselines is publicly available, it often does not support fitness prediction out-of-the-box or, when it does, the codebase does not necessarily provide all the required data processing logic (e.g., pre-processing of MSAs in MSA Transformer) or handle all possible edge cases that may be encountered (e.g., scoring of sequences longer than context size in the ESM suite). Our GitHub repository addresses all of these gaps and provides a consistent interface that will aid in the seamless integration of new baselines as they become available.

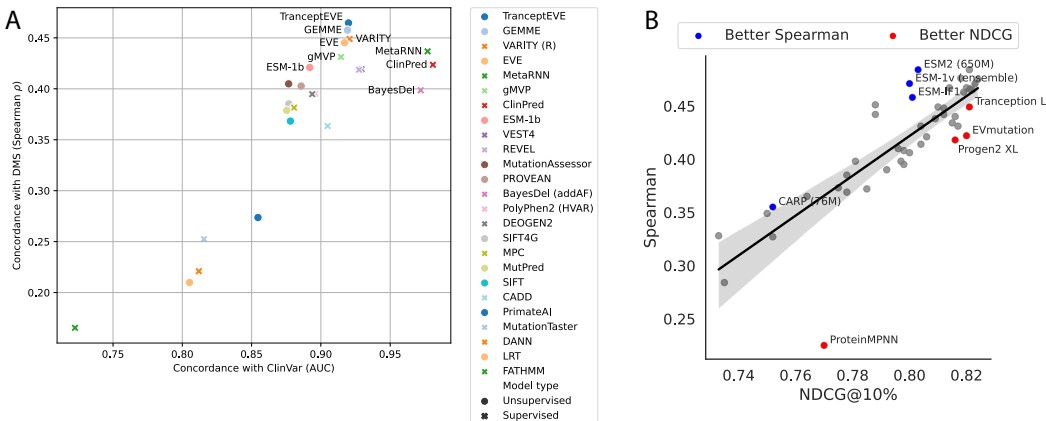

Figure 2: **Comparing baselines across datasets and across performance metrics** (A) Performance estimated against known clinical labels (avg. AUC over genes in ClinVar (x axis)), and DMS assays assessing the clinical effect of variants in humans (avg. Spearman (y axis)). (B) The zero-shot models' median NDCG@10% (x-axis) against median Spearman (y-axis) on the DMS substitutions.

| Model type | Model name | Spearman by DMS type (↑) | | | AUC (↑) |
|---|---|---|---|---|---|
| | | Library | Designed/Natural | All | All |
| Alignment models | HMM | 0.373 | 0.518 | 0.389 | 0.744 |
| | WaveNet | 0.323 | **0.597** | 0.368 | 0.720 |
| | PROVEAN | 0.306 | 0.585 | 0.347 | 0.725 |
| Protein language models | RITA L | 0.443 | 0.519 | 0.457 | 0.773 |
| | ProtGPT2 | 0.185 | 0.128 | 0.191 | 0.620 |
| | ProGen2 M | **0.472** | 0.205 | **0.465** | **0.776** |
| Hybrid models | Tranception M | 0.395 | 0.544 | 0.394 | 0.733 |
| | Tranception L | 0.387 | 0.563 | 0.395 | 0.741 |
| | TranceptEVE M | 0.426 | 0.587 | 0.424 | 0.754 |

Table 4: **Zero-shot indel DMS benchmark** Spearman's rank correlations and AUC between model scores and experimental measurements.

**Processed datasets**   We also make publicly available all processed datasets used in our various benchmarks in a consistent format, including all DMS assays, model scores, ClinVar/gnomAD datasets, predicted 3D structures and Multiple Sequence Alignments required for training and scoring (see Section A.3.3 for more details).

**Website**   Lastly, we developed a user-friendly website in which all benchmarks are accessible, with functionalities to support drill analyses across various dimensions (e.g., mutational depth, taxa) and exporting capabilities.

## 7   Conclusion

ProteinGym addresses the lack of large-scale benchmarks for the robust assessment of models developed for protein design and fitness prediction. It facilitates the direct comparison of methods across several dimensions of interest (e.g., MSA depth, mutational depth, taxa), based on different ground truth datasets (e.g., DMS assays vs Clinical annotations), and in different regimes (e.g., zero-shot vs supervised). We expect the ProteinGym benchmarks and the various data assets we publicly release along with them, to be valuable resources for the Machine Learning and Computational Biology communities, and we plan to continue updating the benchmarks as new assays and baselines become available.

## Acknowledgments and Disclosure of Funding

P.N. was supported by GSK and the UK Engineering and Physical Sciences Research Council (ESPRC ICASE award no.18000077). Y.G. holds a Turing AI Fellowship (Phase 1) at the Alan Turing Institute, which is supported by EPSRC grant reference V030302/1. A.K., P.N., and D.S.M. are supported by a Chan Zuckerberg Initiative Award (Neurodegeneration Challenge Network, CZI2018-191853). S.P., H.S., and D.S.M. are supported by a NIH Transformational Research Award (TR01 1R01CA260415). L.vN. and R.O. gratefully acknowledge funding from CZI NDCN. R.W. is supported by the UK Engineering and Physical Sciences Research Council, as part of the CDT in Health Data Science. We thank the broader Marks Lab, in particular Javier Marchena-Hurtado, for helpful discussions when writing this manuscript. We gratefully acknowledge the compute resources provided by Invitae to train most of the EVE models that we used for the clinical benchmark.

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

# A   Appendix

## A.1   Social Impact

Protein design holds considerable promise for various fields, ranging from medicine to agriculture, and is likely to have a profound social impact. However, the development of such technology introduces several concerns, particularly relating to the dual use of protein fitness and design models. For instance, while beneficial for areas like drug design, these models can also be potentially utilized for harmful purposes such as bio-weapon design. Consider a generative model developed for therapeutic purposes: it typically penalizes predicted toxicity. Yet, the logic of this model could be inverted to instead reward for toxicity [Urbina et al., 2022]. Indeed, any tool or benchmark developed to improve protein design can be manipulated for nefarious objectives. Lastly, protein fitness models will significantly influence the way experiments are conducted. With increased adoption and development of protein design, substantial portions of experimental work can be accelerated, leading to quicker iterations and improved results. Nonetheless, the need for wet lab experimentation remains. These technological advancements will serve to augment, rather than completely supplant, traditional experimental procedures.

Additionally, the American College of Medical Genetics (ACMG) disregards computational prediction of variant effects due to insufficient validation. Consequently, it is essential to create benchmarks using clinical data in order to promote the acceptance of these machine learning methods in medical practice.

## A.2   Limitations

**Deep mutational scans**   While significant efforts have been dedicated to curating and preprocessing a diverse set of deep mutational scans (DMS), the very nature of these scans imposes biases and limitations to this benchmark:

1. **Measurement noise**   Experiments do not have a perfect dynamic range, often imposing a restrictive ceiling and/or floor to the measured response of mutation effects that is not meaningful for protein function and mutation effect prediction. Furthermore, noise is a perennial issue in high-throughput assays, and some assays have poor experimental replicate correlation. Taken together, this means that one cannot expect perfect correlation between experiment and model. Since these considerations affect different proteins to different extents, computing average Spearman correlations across proteins can be misleading.

2. **Bias**   There is additional bias in the types of proteins chosen for deep mutational scans. This can be due both to experimental limitations on which proteins' functions can be assayed (for example, disordered proteins are challenging), and to protein prioritisation considerations (for example, viral and cancer-related proteins are over-represented).

3. **Representativeness**   No assay is fully representative of the impacts of protein changes on the evolutionary fitness of an organism, which typically involves a convolution of molecular functions across changing environments. In fact, many assays target only a single feature such as expression, binding, or enzymatic activity.

4. **Inconsistent processing**   The reported fitness effects from DMS are themselves the result of modeling and analysis of the raw data. The treatment of data is extremely heterogeneous across the community and different analyses can lead to different conclusions on the effect of mutations. For a perfect standardised curation of experimental results one should treat all data with the same approach. We leave this type of analysis for future work.

**Human mutation databases**   ClinVar data has the advantage of covering more proteins than DMS, even if only human proteins involved in disease. But it has several limitations:

1. **Noise**   This dataset, by the very community-based nature of it, is very noisy. Filtering to more stringently curated ClinVar labels, or to more recent labels, improves correlation with predictions from sequence models [Frazer et al., 2021]. Here we decided to keep a reasonable number of clinical labels – a trade-off between quantity and quality.

2. **Bias**   Clinical labels are biased towards classes of proteins that are heavily studied, such as well-known cancer genes, as well as towards European ancestry.

3. **Circularity**   Grimm et al. [2015] details two types of circularity that hinder the evaluation of human variant effect predictors. In a supervised benchmark, there is the potential for data leakage from training to testing, even for different variants in the same protein. Even for our unsupervised benchmark, where models have not trained on clinical labels, there is the potential for another type of circularity: evolutionary conservation is one of the criteria used to classify a variant as benign or pathogenic in ClinVar.

Finally, the current benchmarks are limited to mutations in coding regions. But there are both DMS datasets and clinical labels (although fewer of them) in regulatory regions – for example in UTRs, introns, promoters. This could be an interesting direction of growth for these benchmarks.

## A.3 Datasets

### A.3.1 DMS assays

**Evolution of protein fitness benchmarks based on DMS assays**   As discussed in § 2, our DMS benchmarks build on several prior works that had compiled a growing library of such assays. We summarize their content in Table A1.

| Category | Mut. Type | Metric or Setting | EVmutation | Deep Sequence | ProteinGym v0.1 | ProteinGym v1.0 |
|---|---|---|---|---|---|---|
| DMS | Sub. | Assays (mut.) | 26 (0.1M) | 38 (0.7M) | 87 (1.6M) | 217 (2.4M) |
|  | Ind. | Assays (mut.) | 0 (0k) | 0 (0k) | 7 (270k) | 66 (289k) |
| Clinical | Sub. | Genes (mut.) | 0 | 0 | 0 | 2,525 (63k) |
|  | Ind. | Genes (mut.) | 0 | 0 | 0 | 1,555 (3k) |
| Training regime | Sub. | Zero-shot | ✓ | ✓ | ✓ | ✓ |
|  | Ind. | Zero-shot | - | - | ✓ | ✓ |
|  | Sub. | Supervised | - | - | - | ✓ |
|  | Ind. | Supervised | - | - | - | ✓ |
| Baselines | Sub. | Zero-shot | 5 | 3 | 9 | 42 |
|  | Ind. | Zero-shot | 0 | 0 | 3 | 20 |
|  | Sub. | Supervised | 0 | 0 | 0 | 9 |
|  | Ind. | Supervised | 0 | 0 | 0 | 3 |
| Metrics | - | Zero-shot | 2 | 3 | 3 | 5 |
|  | - | Supervised | 0 | 0 | 0 | 2 |

Table A1: **Evolution of protein fitness benchmarks** ProteinGym v0.1 corresponds to benchmarks in Notin et al. [2022a], while ProteinGym v1.0 corresponds to benchmarks in this paper. The EVmutation benchmark was introduced in Hopf et al. [2017], while the DeepSequence benchmark was developed in Riesselman et al. [2018]. Sub., Ind. and mut. are shorthands for substitutions, indels and mutants respectively.

**Selection and processing**   We focused on several different criteria when determining which DMS assays to include in ProteinGym. These are:

1. The public availability of data
2. The experimental throughput (how many mutations were assayed)
3. The level of noise between experiment replicates
4. The dynamic range of the assay
5. The assay type (selection, enrichment, etc) and whether or not it captures evolutionary constraints.
6. If the assay used amino-acid substitution or indel mutations (no UTR, tRNA, promoter, etc. variants were included).

**Final list of assays**   In-depth metadata about the assays, including the assay type, UniProt ID, MSA start and end positions, mutated positions, and target sequence, is provided under the `reference_files` directory in the codebase. A complete list of included assays is presented at the end of the appendix (See Tables A19 and A20)

**Processing of large thermostability dataset**   A large dataset of thermostability assays of 331 natural domains [Tsuboyama et al., 2023] contributed 65 assays to our list. We processed these assays as follows:

We used the set of non-redundant natural domains (referred to as Dataset #5 in the original paper). After mapping to UniProt IDs for our DMS id naming convention and removing datasets where none of the tested evolutionary models had a Spearman correlation above 0.2 (suggesting that there is inadequate evolutionary fitness signal in the stability assay, preventing meaningful comparisons between models), we were left with 65 thermostability

| Function type | # Assays | | Description |
| --- | --- | --- | --- |
| | Subs | Indels | |
| Activity | 43 | 3 | Assays that directly or indirectly measure a protein's catalytic (or otherwise biochemical) activity |
| Binding | 14 | 0 | Assays that measure the affinity or the degree to which a protein binds its target |
| Expression | 17 | 2 | Assays that measure how much the protein is expressed in a cell |
| Organismal fitness | 77 | 6 | Assays that measure how much changes in the protein affect an organism's growth rate |
| Stability | 66 | 55 | Assays that measure how thermostable a protein is |

Table A2: **DMS assay function types.** The number of substitution and indel assays in each of the 5 function type categories and a general description used to categorize the assays.

assays of short domains (40-72 residues long). For substitutions, there was 99+% coverage of each position with 14-19 mutations per position (and 52 of those datasets with multiples), and for indels there was a deletion, Gly and Ala insertion at every position.

**Classification of DMS assays**   We grouped the substitution DMS assays into five function types: activity, binding, expression, organismal fitness, and stability, assigning each to a primary class such that the classes are non-overlapping. We provide a brief description of each class in Table A2. We took into account multiple factors to delineate the groups as cleanly as possible, most importantly the type of experiment used (e.g., cell growth, cell sorting, biochemical, stability, etc.). Some assay types presented ambiguities that were resolved as follows:

- **Cell growth**   Growth-based assays link the function of their target protein to cellular survival and growth. These assays generally fall under "Organismal fitness", particularly for complementation assays where mutants are tested for their ability to replace the natural function of the wild-type target and allow cell growth. However, in some cases, cell growth is artificially linked to readouts such as enzymatic activity (e.g., SRC_HUMAN [Ahler et al., 2019], MET_HUMAN [Estevam et al., 2023], Q837P4_ENTFA [Meier et al., 2023]), expression (HXK4_HUMAN [Gersing et al., 2023]), or binding (RASK_HUMAN [Weng et al., 2022]). We recategorized assays that were deemed sufficiently distinct from the protein's natural function in the cell.

- **Cell sorting**   Cell sorting, most commonly fluorescence-activated cell sorting (FACS), artificially selects cells that have been fluorescently labeled according to their function. This class of experiments generally falls into "Activity", "Binding", or "Expression", depending on the choice of labeling target. "Expression" assays label the target protein itself, quantifying its expression levels inside the cell or on the cell surface (e.g., NUD15_HUMAN [Suiter et al., 2020], OPSD_HUMAN [Wan et al., 2019]). Since protein abundance and cell surface presentation are dependent on protein stability, these assays are often characterized elsewhere as stability assays (e.g., PRKN_HUMAN [Clausen et al., 2023], PTEN_HUMAN [Matreyek et al., 2021]). "Activity" assays quantify levels of an enzyme's product or reporters of enzymatic activity (e.g., OXDA_RHOTO [Vanella et al., 2023], A0A247D711_LISMN [Stadelmann et al., 2021], PPARG_HUMAN [UK Monogenic Diabetes Consortium et al., 2016]). "Binding" assays quantify levels of a binding partner or another reporter of protein binding (e.g., ACE2_HUMAN [Chan et al., 2020], GCN4_YEAST [Staller et al., 2018]).

**Cross-validation schemes**   As described in § 4.2, we leverage the 3 types of cross-validation schemes (Random, Contiguous and Modulo) introduced in Notin et al. [2023] for the different analyses in the supervised regime. For the Random split, we randomly assigned each mutant to one of 5 folds. The Contiguous scheme is obtained by splitting the sequence in contiguous segments along its length, ensuring the segments are comprised of the same number of positions. We only consider positions mutated, which may not span the entire length of the protein sequence. The Modulo scheme is obtained by assigning positions to each fold using the modulo of the position number by the total number of folds. Therefore, for a 5-fold cross-validation, position 1 is assigned to fold 1, position 2 to fold 2, ..., position 6 to fold 1, etc. Once again, we make sure to only consider mutated positions. We operate a five fold cross-validation for all assays except for assays F7YBW8_MESOW [Aakre et al., 2015] and SPG1_STRSG [Wu et al., 2016], as these contain only 4 mutated positions. Note that multiple mutants generally involve several positions that may not be easily separated into the independent folds as discussed in the Contiguous and Modulo schemes above. Similarly, indels do not lend themselves well to these two cross-validation schemes. Thus, we only keep single mutants for all supervised analyses related to substitutions, and only focus on the Random cross-validation scheme for all indels analyses.

**High-level statistics**  Table A3 describes the size and mutation depth of the indel datasets.

| Dataset | #Datapoints (Benign/Path.) | Mutation Depth (Min/Mean/Max) | Mutation Source |
|---|---|---|---|
| DMS Assays | | | |
| AAV | 24,909 | 1 / 3.57 / 11 | randomization |
| $\beta$-Lac | 4,751 | 1 / 1 / 1 | library |
| Kir2.1 | 10,502 | 1 / 1.2 / 3 | library |
| MtrA | 331 | 8 / 8 / 8 | library |
| PTEN | 314 | 1 / 1 / 1 | library |
| TP53 | 341 | 1 / 1.5 / 2 | library |
| amyloid $\beta$ | 2,354 | 1 / 14 / 39 | library |
| OCT1 | 543 | 1 / 1 / 1 | library |
| Tsuboyama | 14,280 | 1 / 2.7 / 3 | library |
| Assays of Natural and Designed Sequences | | | |
| AAV | 225,998 | 3 / 13.9 / 37 | model-designed |
| CM | 3,074 | 1 / 68.9 / 82 | model-designed |
| HIS3 | 6,102 | 1 / 8.4 / 29 | interpolations between natural sequences |
| Human Variants | | | |
| ClinVar | 3k (1,760 / 839) | 1 / 1.37 / 3 | population variation |

Table A3: **Summary of indel datasets.**

### A.3.2  Clinical datasets

We collect 65k variants from the ClinVar and gnomAD databases (Table A4).

| Dataset | #Proteins | #Variants | #Variants per Protein (Median) |
|---|---|---|---|
| Substitutions | 2,525 | 63k | 6 |
| Indels | 1,555 | 3k | 1 |

Table A4: **Summary of ClinVar human variant datasets.**

For our indel benchmark, detailed in Section 4.1, we focus on short indels, less than or equal to three amino acids, which make up over 80% of in-frame indels in our data. There were insufficient benign annotations for indel clinical variants, so gnomAD common variants (allele frequency > 0.5%) were used as a pseudocontrols.

**ClinVar processing**  The clinical substitutions dataset was obtained following the procedure from EVE [Frazer et al., 2021], detailed in Supplementary Methods Section 3 of that paper (which dataset is downloadable from `https://evemodel.org/download/bulk`), but correcting for mapping errors to GRCh38, which yielded 2,525 proteins and 63k variants, with Pathogenic/Likely Pathogenic/Benign/Likely Benign annotations and at least 1 star of clinical evidence - where assertion criteria is provided by a submitter. As a result, our dataset contains significantly more mutants than the dataset from [Frazer et al., 2021] (42k vs. 63k).

The raw set of inframe indels was obtained from ClinVar on February 6th, 2023, by using the following query:

```
("inframe deletion"[Molecular consequence] OR "inframe indel"[Molecular consequence]
OR "inframe insertion"[Molecular consequence])
```

This query yielded 18407 variants. After filtering out invalid/uncertain amino acids, repeats, remaining frameshift variants, and synonymous/stop codons, 17039 (92.5%) remained.

When filtering for Benign/Pathogenic/Likely Benign/Likely Pathogenic annotations (80%+ of annotations are uncertain significance), and selecting variants in genes with at least one P/LP annotation, and filtering indels up to 3 amino acids, 2090 / 18407 = 11.35% of the original variants remained, 330 benign and 1760 pathogenic. When using gnomAD as the benign pseudocontrols, we only keep the 1760 pathogenic variants from ClinVar.

All the preprocessing code from raw ClinVar data is available in the companion codebase.

**gnomAD processing** The Genome Aggregation Database (gnomAD) [Karczewski et al., 2020] seeks to aggregate genome and exome sequencing data from multiple large-scale sequencing projects, and publishes summary data such as variant allele frequencies in a consistent format. The gnomAD v2.1.1 GRCh38 liftover was downloaded on February 8th 2023 and contains 125,748 exomes and 15,708 genomes. v2 was originally based on the GRCh37 reference sequence and v2.1.1 was lifted over to the GRCh38 reference sequence.

The inframe indels were similarly preprocessed to the ClinVar indels (the preprocessing code from raw data is also available in the repository), yielding 839 common indels up to 3 amino acids in length.

### A.3.3 Access

The following provides more details on the code and data Resources (§ 6) accompanying this paper.

The open-source codebase containing a framework for scoring all the benchmarks is available via our GitHub repository at: `https://github.com/OATML-Markslab/ProteinGym`. Modifications of certain baselines (e.g. scoring of long sequences beyond the context size in the ESM suite, or pre-processing of MSAs in MSA Transformer) are also released, and all of the model predictions can be reproduced using this repository. We also include preprocessing code for the clinical data (ClinVar/gnomAD) and DMS assays for reproducibility.

We developed a user-friendly website, `https://www.proteingym.org` containing a leaderboard, detailed results per assay, as well as drill analyses across various dimensions (e.g mutational depth, taxa).

The DMS assays, model scores, Multiple Sequence Alignments, predicted 3D structures, processed Clin-Var/gnomAD datasets, and raw files before preprocessing can all be downloaded from our servers (see download instructions on our GitHub repository). Some model checkpoints and other files necessary for scoring (for baselines such as profileHMM, PROVEAN) are also available via our servers, although most model checkpoints such as ESM-1v are available from their respective repositories.

### A.3.4 License

The codebase is open source under the MIT license.

## A.4 Baselines

Unless otherwise specified, model scores are calculated by taking the log-ratio of the sequence probabilities between the mutant and wild-type sequences $\log \frac{p(\mathbf{x}_{\mathrm{mut}})}{p(\mathbf{x}_{\mathrm{wt}})}$, following the convention in Hopf et al. [2017].

### A.4.1 Zero-shot baselines

**Alignment-based models**

- **Site-independent Model** We use a site-wise maximum entropy model to infer the contribution of site-specific amino acid constraints without considering explicit epistatic constraints. This model is implemented as referred to in Hopf et al. [2017].

- **HMM** We use the profile hidden Markov model (HMM) implementation in HMMER [Eddy, 2011]. Profile HMMs are frequently used to generate multiple sequence alignments, but also produce log probabilities of sequences that can be used as estimates of fitness for both substitutions and indels [Durbin et al., 1998].

- **EVMutation** EVMutation [Hopf et al., 2017] models pairwise evolutionary couplings between protein sequences using a Potts model (otherwise known as a Markov Random Field).

- **DeepSequence** DeepSequence [Riesselman et al., 2018] uses a VAE architecture to learn higher-order non-linear evolutionary constraints within each protein family. Mutation effect scores are calculated similarly as EVMutation, as the log-ratio between the mutant and wild-type sequence probabilities $\log \frac{p(\mathbf{x}_{\mathrm{mut}}|\theta)}{p(\mathbf{x}_{\mathrm{wt}}|\theta)}$, but using the VAE evidence lower bound (ELBO) as a proxy for $p(\mathbf{x}|\theta)$.

- **WaveNet** We use a previously published dilated convolutional neural network (dilCNN) based on the WaveNet architecture [Shin et al., 2021] as an example of a family-specific sequence decoder capable of handling indels. Due to the expense of training a separate model for each protein, we only evaluate this model against the DMS datasets. Sequence scores are calculated as the difference in (length-normalized) log-likelihoods between the mutant and wild-type sequences.

- **EVE** EVE [Frazer et al., 2021] is a Bayesian VAE model architecture for predicting clinical variant effects. The model includes a Gaussian Mixture Model fitted to the background distribution of mutations, in order to provide interpretable protein-specific pathogenicity scores. We use the ClinVar preprocessing pipeline from EVE, and EVE is also used in TranceptEVE [Notin et al., 2022b].

- **GEMME**    GEMME is the Global Epistatic Model for predicting Mutational Effects. It infers the conservation of combinations of mutations across the entire sequence according to an evolutionary tree [Engelen et al., 2009] and combines it with site-wise frequencies to calculate a combined epistatic sequence score for mutations [Laine et al., 2019]. GEMME intakes multiple sequence alignments of protein families as well as specific mutations to generate scores. To obtain scores, we used the GEMME web-tool hosted at `http://www.lcqb.upmc.fr/GEMME/submit.html` with default parameters.

**Protein language models**    Protein language models are so called because they all use variants of the Transformer [Vaswani et al., 2017] architecture popularised in natural language processing.

- **UniRep**    UniRep [Alley et al., 2019b] trains a Long Short-Term Memory (LSTM) model on UniRef50 [Suzek et al., 2015] sequences. It learns how to internally represent proteins by being trained on next amino acid prediction through minimizing cross-entropy loss. While the core model is trained on unaligned sequences, UniRep can also be fine-tuned on sets of homologous sequences from a given family, retrieved with a MSA. This process is called 'evotuning' and typically leads to stronger fitness prediction performance.

- **ESM**    ESM-1b [Rives et al., 2021] and ESM-1v [Meier et al., 2021] are protein language models with a Transformer encoder architecture similar to BERT [Devlin et al., 2019] and trained with a Masked-Language Modeling (MLM) objective on UniRef50 and UniRef90 respectively. We extend the original ESM codebase for these two models to handle sequences that are longer than the model context window (ie., 1023 amino acids), with the approach described in Brandes et al. [2023] for ESM-1b and in Notin et al. [2022a] for ESM-1v. We predict fitness for ESM models with the masked-marginal approach introduced in Meier et al. [2021], which provides optimal performance on substitutions, but does not support indels.

- **CARP**    CARP [Yang et al., 2023a] is a protein language model trained with a MLM objective on Uniref50. The architecture leverages convolutions instead of self-attention, leading to computational speedups while maitenaning high downstream task performance.

- **RITA**    RITA [Hesslow et al., 2022] is an autoregressive language model akin to GPT2 [Radford et al., 2019], trained on UniRef100 [Suzek et al., 2015]. Four model sizes are available, ranging from 85 million to 1.2 billion parameters. RITA takes unaligned sequences as input, and can score both substitution and indel mutations.

- **ProGen2**    ProGen2 [Nijkamp et al., 2022] is an autoregressive protein language model trained on a mixture of UniRef90 [Suzek et al., 2014] and BFD30 [Steinegger and Söding, 2018]. It follows the standard transformer decoder architecture, and five models of different sizes are available, ranging from 151 million to 6.4 billion parameters. ProGen2 takes unaligned sequences as input, and can score both indel and substitution mutations.

- **VESPA**    VESPA [Marquet et al., 2022] is as Single Amino acid Variant (SAV) effect predictor based on a combination the embeddings from the protein language model ProtT5 [Elnaggar et al., 2021], as well as per-residue conservation predictions.

**Inverse Folding models**    Inverse folding models learn the conditional distribution of sequences that are likely to fold to an input protein structure [Ingraham et al., 2019]. Given that there may not be experimentally solved structures for the target sequence of all DMS assays in ProteinGym, we generate input structures using Alphafold2 (AF2) [Jumper et al., 2021]. The inverse folding model in combination with AF2 encompasses an end-to-end scoring pipeline that only requires a protein sequence to score variants. As the sequence representation size is defined by the size of the input structure, the models we benchmark here can only score substitutions.

- **ProteinMPNN**    ProteinMPNN [Dauparas et al., 2022] takes in a protein backbone structure and featurizes it as a graph where backbone (N,C,C$\alpha$) atoms are nodes and edges are determined via euclidian distance cut-offs. The model uses a message passing neural network (MPNN) [Ingraham et al., 2019] to encode the structure into a latent graph representation. The model then decodes the representation and samples sequences autoregressively.

- **MIF**    The Masked Inverse Folding (MIF) and Masked Inverse Folding with Sequence Transfer (MIF-ST) models [Yang et al., 2023b] are structured-conditioned protein language models trained with a MLM objective. MIF is trained on CATH4.2 [Dawson et al., 2016], and MIF-ST further augments the MIF model with embeddings from CARP (640M).

- **ESM-IF1**    ESM-IF1 [Hsu et al., 2022b] functions similarly to ProteinMPNN but leverages a Geometric Vector Perceptron [Jing et al., 2020] (an equivariant message passing module ideal for coordinate data) as the architecture for the structure encoder and sequence decoder.

**Hybrid models**

- **MSA Transformer** The MSA Transformer [Rao et al., 2021] learns a representation of Multiple Sequence Alignments (MSAs) by training an Axial transformer-based transformer [Ho et al., 2019a] with a MLM objective across a diverse set of 26 million MSAs.

- **Tranception** Tranception [Notin et al., 2022a] combines an autoregressive protein language model with inference-time retrieval from a MSA. We evaluate Tranception Small (85M), Tranception Medium (300M parameters) and Tranception Large (700M parameters) both with and without MSA retrieval. Tranception can score both indel and substitution mutations.

- **TranceptEVE** TranceptEVE augments Tranception with priors for the amino acid distribution at each position based on an ensemble of EVE models for the protein family of interest. The final output log probability is thus a weighted sum of that EVE log prior, the log probability from the autoregressive transformer model in Tranception, as well as site-specific log probabilities obtained from a retrieved MSA (as in the inference-time retrieval procedure described in Tranception). TranceptEVE can score both indels and substitutions.

### A.4.2 Supervised baselines

We leverage the various supervised baselines defined in Notin et al. [2023]:

- **One-hot encoding (OHE) models** OHE baselines take as input a one-hot encoding representation of the amino acid sequence, together with zero-shot fitness predictions obtained with several of the baselines discussed above in Appendix A.4.1. Both are input into a L2-penalized regression, following the approach discussed in [Hsu et al., 2022a];

- **Embeddings models** Embeddings models are based on mean-pooled embeddings from various protein language models introduced above (e.g., Tranception, ESM-1v, MSA Transformer), augmented with zero-shot fitness predictions from the same model. We refer to Notin et al. [2023] for all implementation details;

- **ProteinNPT** ProteinNPT [Notin et al., 2023] is a semi-supervised non-parametric transformer [Kossen et al., 2022] which learns a joint representation of full batches of labeled sequences. It is trained with a hybrid objective consisting of fitness prediction and masked amino acids reconstruction. The model can be used to predict mutation effects for single or multiple properties simultaneously, and sample novel sequences conditioned on label values of interest.

### A.4.3 Clinical baselines

We leverage a set of clinical variant effect predictors from dbNSFP v4.4a [Liu et al., 2011, 2020], which is a database of functional predictions for all possible non-synonymous single-nucleotide variants (nsSNVs) in the human genome.

These models were developed primarily to assess the effects of mutations in humans and are included in clinical benchmarks only:

- **Supervised** The following assays used ClinVar label annotations in their training (or are meta-predictors that contain one or more supervised models): ClinPred [Alirezaie et al., 2018], MetaRNN [Li et al., 2022], BayesDel [Feng, 2017], VEST4 (variant effect scoring tool 4.0) [Carter et al., 2013], REVEL [Ioannidis et al., 2016], VARITY [Wu et al., 2021], gMVP [Zhang et al., 2022], CADD [Rentzsch et al., 2019], PolyPhen2 [Adzhubei et al., 2010], DEOGEN2 [Raimondi et al., 2017], MPC [Samocha et al., 2017], MutationTaster [Schwarz et al., 2010], DANN [Quang et al., 2015], FATHMM[Shihab et al., 2013]

- **Unsupervised** In addition to TranceptEVE, GEMME, EVE and ESM-1b (zero-shot baselines mentioned above), the following unsupervised clinical variant effect predictors were used as baselines: PROVEAN [Choi et al., 2012], SIFT [Ng and Henikoff, 2002], MutationAssessor [Reva et al., 2011], MutPred [Li et al., 2009], PrimateAI [Sundaram et al., 2018], LIST-S2 [Malhis et al., 2020], and LRT [Chun and Fay, 2009].

For ESM-1b, we downloaded precomputed scores from [Brandes and Ntranos, 2023] from a recent study that extended ESM-1b to predict all possible missense variant effects in the human genome [Brandes et al., 2023]. TranceptEVE and EVE models were trained for the subset of 2,525 proteins in the clinical benchmark, and the model weights/scores are provided online for further analysis (See Section A.3.3). GEMME scores were obtained as detailed above. We provide an analysis of performance on clinical datasets vs the subset of assays on human proteins in Fig. 2.

## A.5 Detailed performance results

### A.5.1 DMS substitution benchmarks

**Zero-shot** Table A5 shows the results for our zero-shot DMS substitutions benchmark. We report Spearman's rank correlations and bootstrapped standard error estimates for forty baseline models. Table A6 breaks down our substitution DMS by MSA depth, , Table A7 by function type, Table A8 by taxa, and Table A9 by mutational depth. To compute the final Spearman's rank correlation reported in Table A5, we first average all the assays for a particular function type together, resulting in five average values (one each for Activity, Binding, Expression, Organismal Fitness, and Stability). The average of these five numbers is the final reported value.

**Clustering zero-shot substitution models** We clustered the zero-shot models using hierarchical clustering on the vector of NDCG metrics for each dataset in the DMS substitutions (Fig. A1). We find that models with the same architecture tend to cluster together (e.g., RITA models), however, there are exceptions (e.g., ESM-2 models). We also observe that the alignment-based models tend to cluster together, suggesting that training on the same MSA may promote similar scoring behavior.

**Supervised** Table A10 shows the results for our supervised DMS substitutions benchmark. We report Spearman's rank correlations for 10 baseline models.

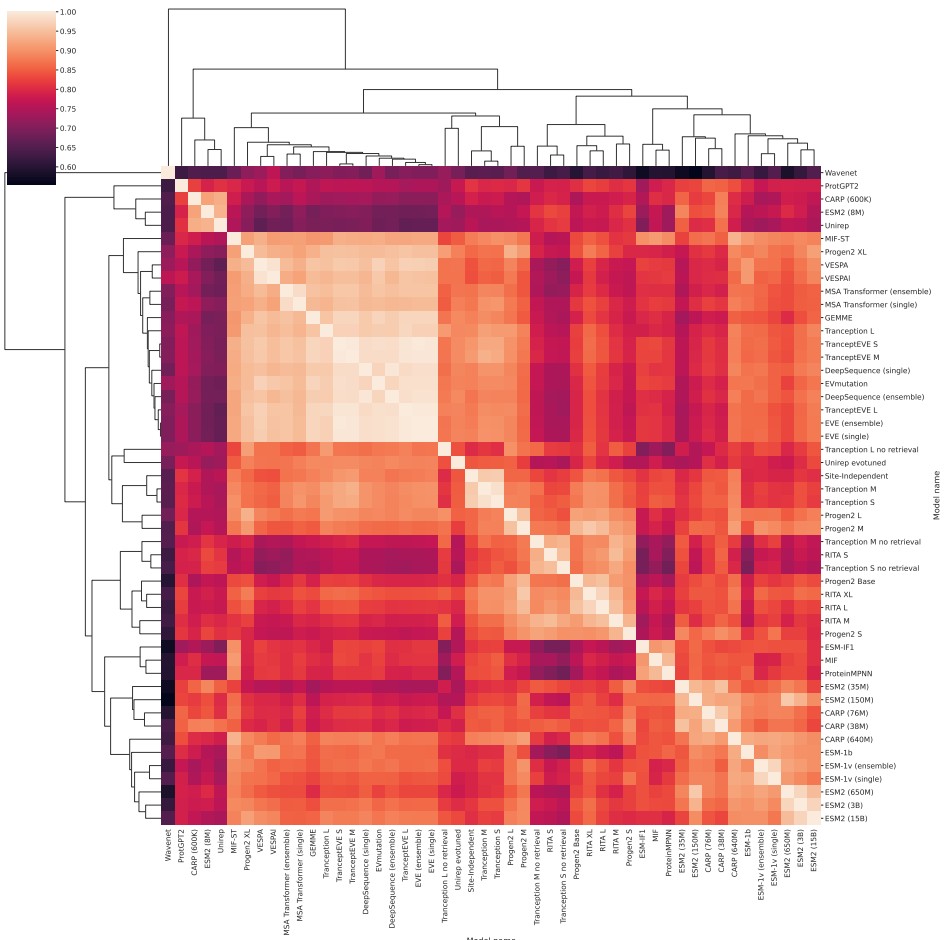

Figure A1: **Hierarchical clustering of zero-shot models by NDCG performance** Heatmap colored by the Pearson correlation of the NDCG@10% values for each DMS assay for each pair of zero-shot models. Lighter color corresponds to higher correlation. The ordering and dendogram were produced by hierarchical clustering of the correlation values.

### A.5.2 DMS indel benchmarks

**Zero-shot**   Table A14 shows the results for our zero-shot DMS indels benchmark, and Table A15 shows Spearman's rank correlations for each indel DMS dataset and model. Figure A2 compares each model's aggregate performance between the Library and Designed DMS sets (numbers provided in Table 4). More detailed performance files are available in the repository.

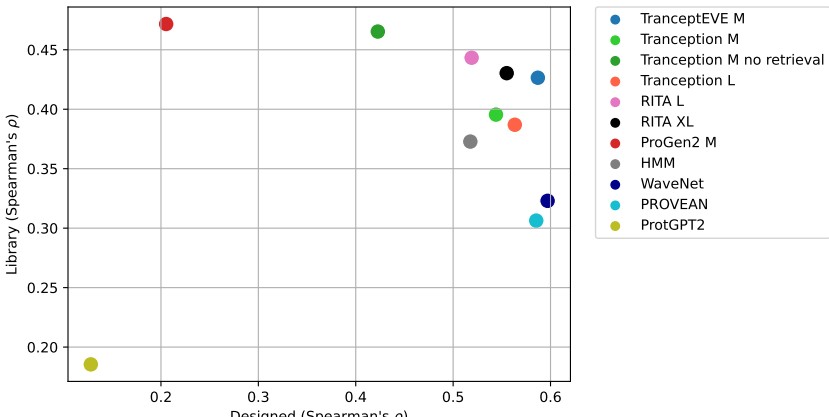

Figure A2: **Performance comparison of indel baselines on different types of assays** Spearman's rank correlation over unbiased libraries vs model-designed sequences biased towards natural sequences.

**Supervised**   Table A16 ranks the performance of each model on the supervised indel DMS benchmark.

### A.5.3 Clinical substitution benchmarks

As discussed in  § 4, since the performance of zero-shot models is on par – or higher – than their supervised counterparts we subsume the Clinical zero-shot and supervised rankings into a combined rankings, available in Table. A17. Although supervised models trained on ClinVar labels (such as ClinPred) perform well on the clinical benchmark, unsupervised models (such as TranceptEVE) provide better performance on the subset of DMS assays assessing the clinical effect of variants in humans, and competitive performance on the clinical benchmark without being subject to the same label biases (see Fig. 2).

### A.5.4 Clinical indel benchmarks

Table A18 shows model performance on the ClinVar datasets, and Figure A3 shows the combined performance on the DMS and ClinVar indel benchmarks.

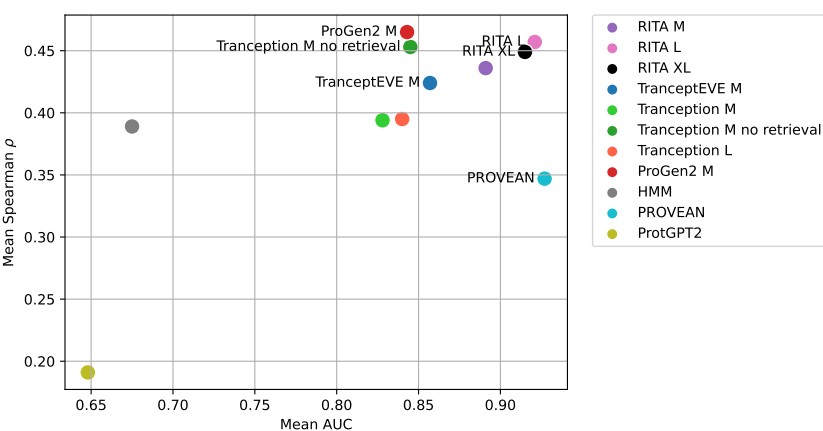

Figure A3: **Performance comparison of indel baselines on the indel benchmarks.** AUC over ClinVar with gnomAD controls (x axis) and Spearman's rank correlation over functional assay benchmarks (y axis).

| Ranking | Model | Type | Spearman | Std. error |
|---------|-------|------|----------|------------|
| 1* | TranceptEVE L | Hybrid model | 0.456 | 0.000 |
| 1* | TranceptEVE M | Hybrid model | 0.455 | 0.004 |
| 1* | GEMME | Alignment-based model | 0.455 | 0.007 |
| 4 | TranceptEVE S | Hybrid model | 0.452 | 0.004 |
| 5 | EVE (ensemble) | Alignment-based model | 0.439 | 0.006 |
| 6 | VESPA | Protein language model | 0.436 | 0.006 |
| 7* | Tranception L | Hybrid model | 0.434 | 0.004 |
| 7* | MSA Transformer (ensemble) | Hybrid model | 0.434 | 0.009 |
| 9 | EVE (single) | Alignment-based model | 0.433 | 0.005 |
| 10 | Tranception M | Hybrid model | 0.427 | 0.005 |
| 11 | ESM-IF1 | Inverse folding model | 0.422 | 0.011 |
| 12 | MSA Transformer (single) | Hybrid model | 0.421 | 0.009 |
| 13 | DeepSequence (ensemble) | Alignment-based model | 0.419 | 0.008 |
| 14 | Tranception S | Hybrid model | 0.418 | 0.006 |
| 15 | ESM2 (650M) | Protein language model | 0.414 | 0.012 |
| 16* | DeepSequence (single) | Alignment-based model | 0.407 | 0.008 |
| 16* | ESM-1v (ensemble) | Protein language model | 0.407 | 0.012 |
| 18 | ESM2 (3B) | Protein language model | 0.406 | 0.011 |
| 19* | MIF-ST | Inverse folding model | 0.401 | 0.010 |
| 19* | ESM2 (15B) | Protein language model | 0.401 | 0.010 |
| 21 | EVmutation | Alignment-based model | 0.395 | 0.006 |
| 22* | ESM-1b | Protein language model | 0.394 | 0.010 |
| 22* | VESPAl | Protein language model | 0.394 | 0.007 |
| 24 | ProGen2 XL | Protein language model | 0.391 | 0.008 |
| 25 | ESM2 (150M) | Protein language model | 0.387 | 0.013 |
| 26 | MIF | Inverse folding model | 0.382 | 0.011 |
| 27 | ProGen2 L | Protein language model | 0.380 | 0.008 |
| 28 | ProGen2 M | Protein language model | 0.379 | 0.008 |
| 29 | ProGen2 Base | Protein language model | 0.378 | 0.009 |
| 30* | Tranception L no retrieval | Protein language model | 0.374 | 0.008 |
| 30* | ESM-1v (single) | Protein language model | 0.374 | 0.013 |
| 32 | WaveNet | Alignment-based model | 0.373 | 0.012 |
| 33 | RITA XL | Protein language model | 0.372 | 0.009 |
| 34 | CARP (640M) | Protein language model | 0.368 | 0.011 |
| 35 | RITA L | Protein language model | 0.365 | 0.009 |
| 36 | Site-Independent | Alignment-based model | 0.359 | 0.010 |
| 37 | RITA M | Protein language model | 0.350 | 0.010 |
| 38 | Tranception M no retrieval | Protein language model | 0.348 | 0.009 |
| 39 | Unirep evotuned | Hybrid model | 0.347 | 0.009 |
| 40 | ProGen2 S | Protein language model | 0.336 | 0.011 |
| 41 | CARP (76M) | Protein language model | 0.328 | 0.012 |
| 42 | ESM2 (35M) | Protein language model | 0.321 | 0.015 |
| 43 | RITA S | Protein language model | 0.304 | 0.011 |
| 44 | Tranception S no retrieval | Protein language model | 0.303 | 0.012 |
| 45 | CARP (38M) | Protein language model | 0.279 | 0.014 |
| 46 | ProteinMPNN | Inverse folding model | 0.258 | 0.011 |
| 47 | ESM2 (8M) | Protein language model | 0.226 | 0.015 |
| 48 | UniRep | Protein language model | 0.190 | 0.016 |
| 49 | ProtGPT2 | Protein language model | 0.188 | 0.011 |
| 50 | CARP (600K) | Protein language model | 0.106 | 0.016 |

Table A5: **ProteinGym - Zero-shot substitution DMS benchmark** Ranking based on corrected average of Spearman's rank correlation between experimental assay measurement and model prediction. The standard error reported corresponds to the non-parametric bootstrap standard error of the difference between the Spearman performance of a given model and that of the best overall model (i.e., TranceptEVE), computed over 10k bootstrap samples from the set of proteins in the ProteinGym DMS substitution benchmark.

| Model type | Model name | Spearman by MSA depth ($\uparrow$) | | | |
|---|---|---|---|---|---|
| | | Low | Medium | High | All |
| Alignment-based | Site-Independent | 0.426 | 0.373 | 0.320 | 0.373 |
| | WaveNet | 0.299 | 0.389 | 0.452 | 0.380 |
| | EVmutation | 0.403 | 0.423 | 0.410 | 0.412 |
| | DeepSequence (ens.) | 0.383 | 0.428 | 0.473 | 0.428 |
| | EVE (ens.) | 0.425 | 0.453 | 0.481 | 0.453 |
| | GEMME | **0.455** | **0.470** | 0.497 | **0.474** |
| Protein language | UniRep | 0.181 | 0.161 | 0.209 | 0.184 |
| | CARP (640M) | 0.314 | 0.375 | 0.428 | 0.372 |
| | ESM-1b | 0.350 | 0.398 | 0.482 | 0.410 |
| | ESM-2 (15B) | 0.357 | 0.414 | 0.473 | 0.415 |
| | RITA XL | 0.315 | 0.382 | 0.412 | 0.370 |
| | ESM-1v (ens.) | 0.326 | 0.418 | 0.502 | 0.415 |
| | ProGen2 XL | 0.354 | 0.405 | 0.444 | 0.401 |
| | VESPA | 0.427 | 0.455 | 0.484 | 0.455 |
| Hybrid | UniRep evotuned | 0.330 | 0.344 | 0.372 | 0.349 |
| | MSA Transformer (ens.) | 0.404 | 0.450 | 0.488 | 0.447 |
| | Tranception L | 0.432 | 0.438 | 0.473 | 0.448 |
| | TranceptEVE L | 0.451 | 0.467 | 0.492 | 0.470 |
| Inverse Folding | ESM-IF1 | 0.300 | 0.431 | **0.544** | 0.425 |
| | MIF-ST | 0.376 | 0.403 | 0.456 | 0.412 |
| | ProteinMPNN | 0.173 | 0.280 | 0.434 | 0.296 |

Table A6: **ProteinGym - Zero-shot substitution DMS benchmark by MSA depth** Average Spearman's rank correlation between model scores and experimental measurements by MSA depth on the ProteinGym substitution benchmark. Alignment depth is measured by the ratio of the effective number of sequences $N_{\text{eff}}$ in the MSA, following Hopf et al. [2017], by the length covered $L$ (Low: $N_{\text{eff}}/L$ <1; Medium: 1< $N_{\text{eff}}/L$ <100; High: $N_{\text{eff}}/L$ >100). The All column is the average across the 3 depths.

| Model type | Model name | Spearman by Function Type (↑) | | | | | |
|---|---|---|---|---|---|---|---|
| | | Activity | Binding | Expression | Organismal Fitness | Stability | All |
| Alignment-based | Site-Independent | 0.369 | 0.345 | 0.351 | 0.382 | 0.358 | 0.361 |
| | WaveNet | 0.379 | 0.325 | 0.350 | 0.365 | 0.449 | 0.374 |
| | EVmutation | 0.440 | 0.322 | 0.382 | 0.411 | 0.430 | 0.397 |
| | DeepSequence (ens.) | 0.455 | 0.368 | 0.396 | 0.413 | 0.476 | 0.422 |
| | EVE (ens.) | 0.464 | **0.394** | 0.406 | 0.447 | 0.491 | 0.440 |
| | GEMME | 0.482 | 0.387 | 0.443 | 0.452 | 0.519 | **0.457** |
| Protein language | UniRep | 0.182 | 0.203 | 0.230 | 0.141 | 0.210 | 0.193 |
| | CARP (640M) | 0.395 | 0.274 | 0.419 | 0.364 | 0.414 | 0.373 |
| | ESM-1b | 0.428 | 0.289 | 0.427 | 0.351 | 0.500 | 0.399 |
| | ESM-2 (15B) | 0.405 | 0.318 | 0.425 | 0.388 | 0.488 | 0.405 |
| | RITA XL | 0.366 | 0.303 | 0.416 | 0.381 | 0.398 | 0.373 |
| | ESM-1v (ens.) | 0.414 | 0.320 | **0.456** | 0.387 | 0.500 | 0.415 |
| | ProGen2 XL | 0.402 | 0.302 | 0.423 | 0.387 | 0.445 | 0.392 |
| | VESPA | 0.468 | 0.365 | 0.410 | 0.440 | 0.500 | 0.437 |
| Hybrid | UniRep evotuned | 0.355 | 0.304 | 0.366 | 0.346 | 0.366 | 0.347 |
| | MSA Transformer (ens.) | 0.469 | 0.343 | 0.439 | 0.421 | 0.495 | 0.433 |
| | Tranception L | 0.465 | 0.351 | 0.455 | 0.436 | 0.471 | 0.436 |
| | TranceptEVE L | **0.487** | 0.381 | **0.456** | **0.460** | 0.500 | **0.457** |
| Inverse Folding | ESM-IF1 | 0.368 | 0.392 | 0.403 | 0.324 | **0.624** | 0.422 |
| | MIF-ST | 0.390 | 0.323 | 0.432 | 0.373 | 0.486 | 0.401 |
| | ProteinMPNN | 0.197 | 0.165 | 0.198 | 0.165 | 0.566 | 0.258 |

Table A7: **ProteinGym - Zero-shot substitution DMS benchmark by function type** Corrected average of Spearman's rank correlation between model scores and experimental measurements on the ProteinGym substitution benchmark, separated into five functional categories (Activity, Binding, Organismal Fitness, Stability and Expression). 'All' is the average of all the categories.

| Model type | Model name | Spearman by Taxa (↑) | | | | |
|---|---|---|---|---|---|---|
| | | Human | Other Eukaryote | Prokaryote | Virus | All |
| Alignment-based | Site-Independent | 0.379 | 0.385 | 0.316 | 0.383 | 0.366 |
| | WaveNet | 0.391 | 0.410 | 0.427 | 0.328 | 0.389 |
| | EVmutation | 0.409 | 0.444 | 0.422 | 0.388 | 0.416 |
| | DeepSequence (ens.) | 0.442 | 0.469 | 0.460 | 0.344 | 0.429 |
| | EVE (ens.) | 0.453 | 0.487 | 0.468 | 0.428 | 0.459 |
| | GEMME | 0.468 | **0.510** | 0.473 | **0.469** | **0.480** |
| Protein language | UniRep | 0.213 | 0.219 | 0.165 | 0.057 | 0.164 |
| | CARP (640M) | 0.416 | 0.386 | 0.390 | 0.273 | 0.366 |
| | ESM-1b | 0.434 | 0.475 | 0.455 | 0.241 | 0.401 |
| | ESM-2 (15B) | 0.431 | 0.449 | 0.459 | 0.313 | 0.413 |
| | RITA XL | 0.394 | 0.384 | 0.353 | 0.402 | 0.383 |
| | ESM-1v (ens.) | 0.458 | 0.446 | 0.454 | 0.289 | 0.412 |
| | ProGen2 XL | 0.384 | 0.442 | 0.439 | 0.391 | 0.414 |
| | VESPA | 0.438 | 0.492 | 0.490 | 0.432 | 0.463 |
| Hybrid | UniRep evotuned | 0.355 | 0.363 | 0.346 | 0.349 | 0.353 |
| | MSA Transformer (ens.) | 0.437 | 0.505 | 0.463 | 0.414 | 0.455 |
| | Tranception L | 0.453 | 0.483 | 0.431 | 0.432 | 0.450 |
| | TranceptEVE L | **0.471** | 0.498 | 0.473 | 0.453 | 0.474 |
| Inverse Folding | ESM-IF1 | 0.415 | 0.497 | **0.507** | 0.374 | 0.448 |
| | MIF-ST | 0.404 | 0.415 | 0.463 | 0.396 | 0.420 |
| | ProteinMPNN | 0.282 | 0.395 | 0.354 | 0.248 | 0.320 |

Table A8: **ProteinGym - Zero-shot substitution DMS benchmark by taxa** Average Spearman's rank correlation between model scores and experimental measurements on the ProteinGym substitution benchmark, separated by taxon. 'All' is the average across the taxa.

| Model type | Model name | Spearman by Mutational Depth (↑) | | | | | |
|---|---|---|---|---|---|---|---|
| | | 1 | 2 | 3 | 4 | 5+ | All |
| Alignment-based | Site-Independent | 0.336 | 0.235 | 0.226 | 0.267 | 0.350 | 0.283 |
| | WaveNet | 0.357 | 0.204 | 0.250 | 0.217 | 0.293 | 0.264 |
| | EVmutation | 0.376 | 0.274 | 0.324 | 0.301 | 0.394 | 0.334 |
| | DeepSequence (ens.) | 0.405 | 0.264 | 0.313 | 0.309 | 0.378 | 0.334 |
| | EVE (ens.) | 0.428 | 0.273 | 0.308 | 0.298 | 0.355 | 0.332 |
| | GEMME | **0.447** | 0.274 | 0.321 | 0.324 | **0.414** | 0.356 |
| Protein language | UniRep | 0.175 | 0.071 | 0.111 | 0.141 | 0.191 | 0.138 |
| | CARP (640M) | 0.390 | 0.213 | 0.187 | 0.164 | 0.162 | 0.223 |
| | ESM-1b | 0.384 | 0.227 | 0.187 | 0.149 | 0.270 | 0.243 |
| | ESM-2 (15B) | 0.407 | 0.204 | 0.239 | 0.172 | 0.234 | 0.251 |
| | RITA XL | 0.356 | 0.139 | 0.136 | 0.154 | 0.233 | 0.204 |
| | ESM-1v (ens.) | 0.403 | 0.221 | 0.186 | 0.151 | 0.203 | 0.233 |
| | ProGen2 XL | 0.385 | 0.184 | 0.280 | 0.219 | 0.280 | 0.270 |
| | VESPA | 0.434 | 0.183 | 0.357 | 0.302 | 0.328 | 0.321 |
| Hybrid | UniRep evotuned | 0.319 | 0.154 | 0.250 | 0.226 | 0.294 | 0.249 |
| | MSA Transformer (ens.) | 0.426 | 0.238 | **0.384** | **0.366** | 0.408 | **0.364** |
| | Tranception L | 0.423 | 0.258 | 0.352 | 0.318 | 0.387 | 0.348 |
| | TranceptEVE L | 0.446 | 0.280 | 0.350 | 0.320 | 0.382 | 0.356 |
| Inverse Folding | ESM-IF1 | 0.439 | **0.345** | 0.290 | 0.289 | 0.358 | 0.344 |
| | MIF-ST | 0.430 | 0.265 | 0.334 | 0.298 | 0.298 | 0.325 |
| | ProteinMPNN | 0.292 | 0.257 | 0.171 | 0.186 | 0.278 | 0.237 |

Table A9: **ProteinGym - Zero-shot substitution DMS benchmark by mutational depth** Spearman's rank correlation between model scores and experimental measurements on the ProteinGym substitution benchmark, separated by mutational depths of 1,2,3,4, and 5 or more. The All column is the average across the 5 depths.

| Ranking | Model name | Model type | Spearman |
|---|---|---|---|
| 1 | ProteinNPT | NPT | 0.613 |
| 2 | Tranception | Embeddings | 0.571 |
| 3 | MSA Transformer | Embeddings | 0.568 |
| 4 | ESM-1v | Embeddings | 0.542 |
| 5 | TranceptEVE | OHE | 0.477 |
| 6 | Tranception | OHE | 0.458 |
| 7 | MSAT | OHE | 0.453 |
| 8 | DeepSequence | OHE | 0.440 |
| 9 | ESM-1v | OHE | 0.417 |
| 10 | OHE w/o augmentation | OHE | 0.224 |

Table A10: **ProteinGym - Supervised substitution DMS benchmark** Ranking based on corrected average of Spearman's rank correlation between experimental assay measurement and model prediction.

| Model type | Model name | Spearman by MSA depth (↑) | | | |
|---|---|---|---|---|---|
| | | Low | Medium | High | All |
| NPT | ProteinNPT | **0.701** | **0.587** | **0.608** | **0.632** |
| Embeddings | Tranception | 0.621 | 0.556 | 0.561 | 0.579 |
| | MSAT | 0.685 | 0.518 | 0.567 | 0.590 |
| | ESM-1v | 0.653 | 0.465 | 0.541 | 0.553 |
| One-hot encoding | TranceptEVE | 0.503 | 0.483 | 0.468 | 0.485 |
| | Tranception | 0.490 | 0.455 | 0.445 | 0.463 |
| | MSAT | 0.500 | 0.441 | 0.448 | 0.463 |
| | DeepSequence | 0.482 | 0.422 | 0.426 | 0.443 |
| | ESM-1v | 0.496 | 0.338 | 0.400 | 0.411 |
| | No Augmentation | 0.246 | 0.204 | 0.227 | 0.226 |

Table A11: **Supervised substitution DMS benchmark by MSA depth** Average Spearman's rank correlation between model scores and experimental measurements by MSA depth on the ProteinGym substitution benchmark. Alignment depth is measured by the ratio of the effective number of sequences $N_{\text{eff}}$ in the MSA, following Hopf et al. [2017], by the length covered $L$ (Low: $N_{\text{eff}}/L <1$; Medium: $1< N_{\text{eff}}/L <100$; High: $N_{\text{eff}}/L >100$)

| Model type | Model name | Spearman by Function Type (↑) | | | | | |
|---|---|---|---|---|---|---|---|
| | | Activity | Binding | Expression | Organismal Fitness | Stability | All |
| NPT | ProteinNPT | **0.577** | **0.536** | **0.637** | **0.545** | **0.772** | **0.613** |
| Embeddings | Tranception | 0.520 | 0.529 | 0.613 | 0.519 | 0.674 | 0.571 |
| | MSAT | 0.547 | 0.470 | 0.584 | 0.493 | 0.749 | 0.569 |
| | ESM-1v | 0.487 | 0.450 | 0.587 | 0.468 | 0.717 | 0.542 |
| One-hot encoding | TranceptEVE | 0.502 | 0.444 | 0.476 | 0.470 | 0.493 | 0.477 |
| | Tranception | 0.475 | 0.416 | 0.476 | 0.448 | 0.473 | 0.458 |
| | MSAT | 0.480 | 0.393 | 0.463 | 0.437 | 0.491 | 0.453 |
| | DeepSequence | 0.467 | 0.418 | 0.424 | 0.422 | 0.471 | 0.440 |
| | ESM-1v | 0.421 | 0.363 | 0.452 | 0.383 | 0.463 | 0.416 |
| | No Augmentation | 0.213 | 0.212 | 0.226 | 0.194 | 0.273 | 0.224 |

Table A12: **Supervised substitution DMS benchmark by function type** Average Spearman's rank correlation between supervised model scores and experimental measurements on the ProteinGym substitution benchmark, separated into five functional categories. Assays are split into one of Activity, Binding, Organismal Fitness, Stability and Expression. The All column is the average of all the categories

| Model type | Model name | Spearman by Taxa (↑) | | | | |
|---|---|---|---|---|---|---|
| | | Human | Other Eukaryote | Prokaryote | Virus | All |
| NPT | ProteinNPT | **0.649** | **0.628** | **0.668** | **0.580** | **0.631** |
| Embeddings | Tranception | 0.569 | 0.582 | 0.594 | 0.568 | 0.578 |
| | MSAT | 0.634 | 0.579 | 0.648 | 0.521 | 0.596 |
| | ESM-1v | 0.565 | 0.579 | 0.617 | 0.433 | 0.548 |
| One-hot encoding | TranceptEVE | 0.481 | 0.490 | 0.475 | 0.478 | 0.481 |
| | Tranception | 0.457 | 0.472 | 0.453 | 0.456 | 0.460 |
| | MSAT | 0.482 | 0.459 | 0.468 | 0.448 | 0.464 |
| | DeepSequence | 0.451 | 0.460 | 0.455 | 0.383 | 0.437 |
| | ESM-1v | 0.426 | 0.444 | 0.452 | 0.292 | 0.404 |
| | No Augmentation | 0.236 | 0.217 | 0.233 | 0.238 | 0.231 |

Table A13: **Supervised substitution DMS benchmark by taxa** Average Spearman's rank correlation between model scores and experimental measurements on the ProteinGym substitution benchmark, separated into four taxon categories. Assays are split into one of Human, Prokaryote, Other Eukaryote, or Virus. The All column is the average across the categories.

| Ranking | Model | Type | Spearman | Std. error |
|---|---|---|---|---|
| 1 | ProGen2 M | Protein language model | 0.465 | 0.000 |
| 2 | ProGen2 Base | Protein language model | 0.464 | 0.010 |
| 3 | RITA L | Protein language model | 0.457 | 0.034 |
| 4 | Tranception M no retrieval | Protein language model | 0.453 | 0.036 |
| 5* | RITA XL | Protein language model | 0.449 | 0.037 |
| 5* | ProGen2 L | Protein language model | 0.449 | 0.011 |
| 7 | Tranception L no retrieval | Protein language model | 0.437 | 0.041 |
| 8 | RITA M | Protein language model | 0.436 | 0.030 |
| 9 | ProGen2 XL | Protein language model | 0.431 | 0.035 |
| 10* | TranceptEVE M | Hybrid model | 0.424 | 0.045 |
| 10* | ProGen2 S | Protein language model | 0.424 | 0.025 |
| 12 | TranceptEVE L | Hybrid model | 0.412 | 0.046 |
| 13 | Tranception S no retrieval | Protein language model | 0.410 | 0.036 |
| 14 | RITA S | Protein language model | 0.397 | 0.032 |
| 15 | Tranception L | Hybrid model | 0.395 | 0.043 |
| 16 | Tranception M | Hybrid model | 0.394 | 0.047 |
| 17 | HMM | Alignment-based model | 0.389 | 0.045 |
| 18 | WaveNet | Alignment-based model | 0.368 | 0.067 |
| 19 | TranceptEVE S | Hybrid model | 0.357 | 0.049 |
| 20 | PROVEAN | Alignment-based model | 0.347 | 0.046 |
| 21 | Tranception S | Hybrid model | 0.340 | 0.053 |
| 22 | ProtGPT2 | Protein language model | 0.191 | 0.053 |
| 23 | UniRep | Protein language model | 0.169 | 0.060 |

Table A14: **ProteinGym - Zero-shot indel DMS benchmark** Ranking based on corrected average of Spearman's rank correlation between experimental assay measurement and model prediction. The standard error reported corresponds to the non-parametric bootstrap standard error of the difference between the Spearman performance of a given model and that of the best overall model (ie., TranceptEVE), computed over 10k bootstrap samples from the set of proteins in the ProteinGym DMS indel benchmark.

| Dataset | Tranception | | | | TranceptEVE | | ProGen2 | | RITA | WN. | PRO. | HMM |
|---|---|---|---|---|---|---|---|---|---|---|---|---|
| | M | L | M+Ret | L+Ret | M | L | M | XL | L | | | |
| DMS Assays | | | | | | | | | | | | |
| amyloid $\beta$ | 0.466 | 0.416 | 0.442 | 0.444 | 0.459 | 0.466 | 0.478 | 0.462 | 0.438 | **0.512** | 0.381 | -0.207 |
| $\beta$-Lac | 0.365 | 0.344 | 0.379 | 0.296 | 0.401 | 0.342 | **0.619** | 0.409 | 0.334 | 0.437 | 0.385 | 0.347 |
| AAV | 0.371 | 0.338 | 0.126 | 0.210 | **0.419** | 0.416 | -0.100 | 0.167 | 0.103 | -0.007 | 0.177 | 0.057 |
| Kir2.1 | 0.437 | 0.440 | 0.412 | 0.391 | **0.444** | 0.431 | 0.432 | 0.387 | 0.383 | 0.408 | 0.386 | 0.368 |
| TP53 | 0.536 | 0.362 | **0.579** | 0.395 | 0.560 | 0.399 | 0.428 | 0.354 | 0.383 | 0.031 | 0.273 | 0.482 |
| PTEN | 0.678 | 0.546 | 0.700 | 0.563 | **0.708** | 0.602 | 0.552 | 0.402 | 0.504 | 0.697 | 0.237 | 0.668 |
| MtrA | 0.612 | 0.395 | **0.615** | 0.375 | 0.562 | 0.374 | 0.403 | 0.348 | 0.380 | 0.244 | 0.278 | 0.472 |
| OCT1 | 0.379 | 0.458 | 0.447 | 0.466 | 0.442 | 0.453 | **0.546** | 0.383 | 0.542 | 0.087 | 0.301 | 0.280 |
| Tsuboyama | 0.169 | 0.234 | 0.434 | 0.463 | 0.198 | 0.265 | 0.511 | **0.533** | 0.489 | 0.475 | 0.333 | 0.364 |
| Assays of Natural and Designed Sequences | | | | | | | | | | | | |
| AAV | 0.677 | 0.709 | 0.362 | 0.691 | 0.726 | **0.736** | -0.466 | 0.492 | 0.543 | 0.666 | 0.683 | 0.607 |
| CM | 0.344 | 0.326 | 0.219 | 0.223 | 0.357 | 0.340 | 0.380 | 0.379 | 0.337 | **0.438** | 0.372 | 0.398 |
| HIS3 | 0.611 | 0.655 | 0.687 | 0.707 | 0.678 | 0.695 | 0.702 | **0.713** | 0.677 | 0.687 | 0.701 | 0.548 |

Table A15: **Spearman's rank correlation between model scores and individual deep mutational scans of indels.** WN and PRO are shorthands for the WaveNet and PROVEAN models respectively.

| Ranking | Model | Type | Spearman |
|---|---|---|---|
| 1 | ESM-1v | Embeddings | 0.752 |
| 2 | Tranception | Embeddings | 0.735 |
| 3 | MSAT | Embeddings | 0.689 |

Table A16: **ProteinGym - Supervised indel DMS benchmark** Ranking based on corrected average of Spearman's rank correlation between experimental assay measurement and model prediction.

| Ranking | Model | Type | AUC |
|---|---|---|---|
| 1 | ClinPred | Supervised | 0.981 |
| 2 | MetaRNN | Supervised | 0.977 |
| 3 | BayesDel (addAF) | Supervised | 0.972 |
| 4 | VEST4 | Supervised | 0.929 |
| 5 | REVEL | Supervised | 0.928 |
| 6 | BayesDel (noAF) | Supervised | 0.925 |
| 7 | VARITY (R) | Supervised | 0.921 |
| 8 | TranceptEVE | Unsupervised | 0.920 |
| 9 | GEMME | Unsupervised | 0.919 |
| 10 | VARITY (ER) | Supervised | 0.918 |
| 11 | EVE | Unsupervised | 0.917 |
| 12 | gMVP | Supervised | 0.914 |
| 13 | CADD | Supervised | 0.905 |
| 14 | PolyPhen2 (HVAR) | Supervised | 0.896 |
| 15 | DEOGEN2 | Supervised | 0.894 |
| 16 | ESM-1b | Unsupervised | 0.892 |
| 17 | PROVEAN | Unsupervised | 0.886 |
| 18 | MPC | Supervised | 0.881 |
| 19 | PolyPhen2 (HDIV) | Supervised | 0.879 |
| 20 | SIFT | Unsupervised | 0.878 |
| 21 | SIFT4G | Unsupervised | 0.877 |
| 22 | MutationAssessor | Unsupervised | 0.877 |
| 23 | MutPred | Unsupervised | 0.875 |
| 24 | PrimateAI | Unsupervised | 0.855 |
| 25 | LIST-S2 | Unsupervised | 0.842 |
| 26 | MutationTaster | Supervised | 0.816 |
| 27 | DANN | Supervised | 0.812 |
| 28 | LRT | Unsupervised | 0.805 |
| 29 | FATHMM | Supervised | 0.723 |

Table A17: **ProteinGym - Clinical substitution benchmark** Ranking based on AUROC between model prediction and ClinVar benign/pathogenic annotation.

| Model Type | Model Name | AUROC(↑) | AUPRC (↑) |
|---|---|---|---|
| Alignment-based models | HMM | 0.679 | 0.775 |
| | PROVEAN | **0.926** | 0.947 |
| | WaveNet | – | – |
| Protein language models | UniRep | 0.395 | 0.600 |
| | RITA XL | 0.923 | **0.954** |
| | ProGen2 XL | 0.846 | 0.889 |
| | Tranception L (no retrieval) | 0.877 | 0.938 |
| | Tranception M (no retrieval) | 0.858 | 0.929 |
| | ProtGPT2 | 0.655 | 0.779 |
| Hybrid models | Tranception L | 0.857 | 0.920 |
| | Tranception M | 0.844 | 0.909 |
| | TranceptEVE | 0.857 | 0.916 |

Table A18: **ProteinGym - Clinical indels benchmark** Results for indel-compatible baselines on our ClinVar/gnomAD indel benchmark. AUPRC is area under the precision recall curve, and AUROC is area under the receiver-operating characteristic curve. Bold denotes best method, with the runner-up underlined.

Table A19: **List of substitution datasets** See the reference file in the GitHub repo
for other info (UniProt ID, taxon, DOI, and more assay details).

| Dataset | Reference |
| --- | --- |
| $\beta$-Lactamase | Jacquier et al. [2013] |
| $\beta$-Lactamase | Stiffler et al. [2015] |
| $\beta$-Lactamase | Firnberg et al. [2014] |
| $\beta$-Lactamase | Deng et al. [2012] |
| $\beta$-Lactamase VIM-2 | Chen et al. [2020] |
| $\beta$-Glucosidase | Romero et al. [2015] |
| AAV | Sinai et al. [2021] |
| ACE2 | Chan et al. [2020] |
| ADRB2 | Jones et al. [2020] |
| APH(3')II, neo | Melnikov et al. [2014] |
| APP | Seuma et al. [2021] |
| Activation-induced deaminase | Gajula et al. [2014] |
| Aliphatic amidase | Wrenbeck et al. [2017] |
| Alpha-synuclein | Newberry et al. [2020] |
| Amyloid $\beta$ | Gray et al. [2019] |
| Amyloid $\beta$ | Seuma et al. [2022] |
| Ancestral spleen tyrosine kinase | Hobbs et al. [2022] |
| Anti-CRISPR protein AcrIIA4 | Stadelmann et al. [2021] |
| Antitoxin ParD3 | Ding et al. [2023] |
| Antitoxin ParD3 | Aakre et al. [2015] |
| Arrestin-1 | Ostermaier et al. [2014] |
| BRCA1 | Findlay et al. [2018] |
| BRCA2 | Erwood et al. [2022] |
| CALM1 | Weile et al. [2017] |
| CARD11 | Meitlis et al. [2020] |
| CASP3 | Roychowdhury and Romero [2022] |
| CASP7 | Roychowdhury and Romero [2022] |
| CBS (cystathionine beta-synthase) | Sun et al. [2020] |
| CCR5 | Gill et al. [2023] |
| CD19 | Klesmith et al. [2019] |
| CVB3 capsid | Mattenberger et al. [2021] |
| CXCR4 | Gill et al. [2023] |
| Chalcone synthase | Wrenbeck et al. [2019] |
| Cytochrome P450 2C9 | Amorosi et al. [2021] |
| Cytochrome P450 2C9 | Amorosi et al. [2021] |
| D-amino acid oxidase | Vanella et al. [2023] |
| DHFR reductase | Nguyen et al. [2023a] |
| DHFR reductase | Thompson et al. [2020] |
| DNA methylase HaeIII | Rockah-Shmuel et al. [2015] |
| Dengue virus NS5 | Suphatrakul et al. [2023] |
| Dlg4, (PSD95_PDZ3) | McLaughlin et al. [2012] |
| EfrC | Meier et al. [2023] |
| EfrD | Meier et al. [2023] |
| EnvZ | Ghose et al. [2023] |
| ErbB2 membrane domain | Elazar et al. [2016] |
| EstA | Nutschel et al. [2020] |
| GAL4 | Kitzman et al. [2015] |
| GB1 | Wu et al. [2016] |
| GB1 | Olson et al. [2014] |
| GDI1 | Silverstein et al. [2021] |
| GFP | Sarkisyan et al. [2016] |
| GMR (aacC1) | Dandage et al. [2018] |
| GRB2-SH3 | Faure et al. [2022] |
| Gcn4 | Staller et al. [2018] |
| Glucokinase regulatory protein | Gersing et al. [2023] |
| Glucokinase regulatory protein | Gersing et al. [2022] |
| Glycophorin A membrane domain | Elazar et al. [2016] |
| Green fluorescent protein amacGFP | Gonzalez Somermeyer et al. [2022] |
| Green fluorescent protein cgreGFP | Gonzalez Somermeyer et al. [2022] |
| Green fluorescent protein ppluGFP2 | Gonzalez Somermeyer et al. [2022] |

| Dataset | Reference |
|---|---|
| HIV env | Duenas-Decamp et al. [2016] |
| HIV env | Haddox et al. [2018] |
| HIV env (BF520) | Haddox et al. [2018] |
| HIV env (BG505) | Haddox et al. [2018] |
| HIV rev | Fernandes et al. [2016] |
| HIV tat | Fernandes et al. [2016] |
| HMG-CoA reductase | Jiang [2019] |
| HRAS | Bandaru et al. [2017] |
| HSP82 | Flynn et al. [2020] |
| HSP82 | Mishra et al. [2016] |
| Hsp90 | Hietpas et al. [2011] |
| Hydroxymethylbilane synthase | van Loggerenberg et al. [2023] |
| IGP dehydratase (HIS3) | Pokusaeva et al. [2019] |
| InfA | Kelsic et al. [2016] |
| Influenza H3N2 neuraminidase | Lei et al. [2023] |
| Influenza M1 matrix protein | Hom et al. [2019] |
| Influenza RNA polymerase PB1 | Li et al. [2023] |
| Influenza hemagglutinin | Thyagarajan and Bloom [2014] |
| Influenza hemagglutinin | Doud and Bloom [2016] |
| Influenza hemagglutinin | Wu et al. [2014] |
| Influenza hemagglutinin | Lee et al. [2018] |
| Influenza neuraminidase | Jiang et al. [2016] |
| Influenza nucleoprotein | Bloom [2014] |
| Influenza nucleoprotein | Doud et al. [2015] |
| Influenza nucleoprotein | Doud and Bloom [2016] |
| Influenza polymerase acidic protein | Wu et al. [2015] |
| Influenza polymerase basic protein 2 | Soh et al. [2019] |
| KCNE1 | Muhammad et al. [2023] |
| KCNH2 | Kozek et al. [2020] |
| KCNJ2 | Coyote-Maestas et al. [2022] |
| KRAS | Weng et al. [2022] |
| KRAS | Ursu et al. [2022] |
| L-selectin | Elazar et al. [2016] |
| LGK (levoglucosan kinase) | Wrenbeck et al. [2019] |
| LGK (levoglucosan kinase) | Klesmith et al. [2015] |
| LamB | Andrews and Fields [2020] |
| Leucine-rich repeat protein SHOC-2 | Kwon et al. [2022] |
| MAPK1 | Brenan et al. [2016] |
| MET kinase | Estevam et al. [2023] |
| MPL | Bridgford et al. [2020] |
| MSH2 | Jia et al. [2021] |
| MTHFR reductase | Weile et al. [2021] |
| MlaC | MacRae et al. [2023] |
| NPC intracellular cholesterol transporter | Erwood et al. [2022] |
| NPC intracellular cholesterol transporter | Erwood et al. [2022] |
| NS5A | Qi et al. [2014] |
| NUDT15 | Suiter et al. [2020] |
| OCT1 (SLC22A1) | Yee et al. [2023] |
| Ornithine transcarbamylase | Lo et al. [2023] |
| p53 | Giacomelli et al. [2018] |
| p53 | Kotler et al. [2018] |
| PAB1 | Melamed et al. [2013] |
| PPARG | UK Monogenic Diabetes Consortium et al. [2016] |
| PSD95-PDZ3 | Faure et al. [2022] |
| PTEN | Matreyek et al. [2021] |
| PTEN | Mighell et al. [2018] |
| Parkin | Clausen et al. [2023] |
| Phosphoserine aminotransferase | Xie et al. [2023] |
| Phototropin | Chen et al. [2023] |
| Pilin (PilE) | Kennouche et al. [2019] |
| Plasminogen activator inhibitor-1 | Huttinger et al. [2021] |

Table A19: (continued)

| Dataset | Reference |
|---|---|
| Protein phosphatase 1D | Miller et al. [2022] |
| RAF oncogene | Zinkus-Boltz et al. [2019] |
| RNAse III (rnc) | Weeks and Ostermeier [2023] |
| Rhodopsin | Wan et al. [2019] |
| SARS-CoV-2 Mpro | Flynn et al. [2022] |
| SARS-CoV-2 Spike RBD | Tan et al. [2023] |
| SARS-CoV2 Spike RBD | Starr et al. [2020] |
| SCN5A | Glazer et al. [2020] |
| SOX17 | Veerapandian et al. [2018] |
| SOX2 | Veerapandian et al. [2018] |
| SRC | Ahler et al. [2019] |
| SUMO-conjugating enzyme UBC9 | Weile et al. [2017] |
| Small ubiquitin-related modifier 1 | Weile et al. [2017] |
| Sodium-dependent serotonin transporter | Young et al. [2021] |
| Src | Chakraborty et al. [2021] |
| Src | Nguyen et al. [2023b] |
| Streptococcus pyogenes Cas9 | Spencer and Zhang [2017] |
| TARDBP | Bolognesi et al. [2019] |
| TIM Barrel (S. solfataricus) | Chan et al. [2017] |
| TIM Barrel (T. maritima) | Chan et al. [2017] |
| TIM Barrel (T. thermophilus) | Chan et al. [2017] |
| Thiamin pyrophosphokinase 1 | Weile et al. [2017] |
| Thiopurine S-methyltransferase (TPMT) | Matreyek et al. [2018] |
| Toxin CcdB | Tripathi et al. [2016] |
| Toxin CcdB | Adkar et al. [2012] |
| Tsuboyama multi-DMS | Tsuboyama et al. [2023] |
| Ube4b | Starita et al. [2013] |
| Ubiquitin | Roscoe et al. [2013] |
| Ubiquitin | Roscoe and Bolon [2014] |
| Ubiquitin | Mavor et al. [2016] |
| VKORC1 | Chiasson et al. [2020] |
| VKORC1 | Chiasson et al. [2020] |
| YAP1 | Araya et al. [2012] |
| Zika virus env | Sourisseau et al. [2019] |

| Dataset | Reference |
|---|---|
| $\beta$-Lactamase | Gonzalez et al. [2019] |
| AAV | Sinai et al. [2021] |
| Chorismate mutase (CM) | Russ et al. [2020] |
| IGP dehydratase (HIS3) | Pokusaeva et al. [2019] |
| Kir2.1 | Macdonald et al. [2023] |
| MtrA | Campbell et al. [2022] |
| p53 | Kotler et al. [2018] |
| PTEN phosphatase | Mighell et al. [2018] |
| amyloid $\beta$ | Seuma et al. [2022] |
| OCT1 (SLC22A1) | Yee et al. [2023] |
| Tsuboyama multi-DMS | Tsuboyama et al. [2023] |

Table A20: **List of indel datasets.**

