# OpenReview forum: "ProteinGym: Large-Scale Benchmarks for Protein Fitness Prediction and Design"
_NeurIPS.cc/2023/Track/Datasets_and_Benchmarks — NeurIPS 2023 Datasets and Benchmarks Poster_

### Official Review · Reviewer_vaLL · 2023-07-03
**ProteinGym: Large-Scale Benchmarks for Protein Design and Fitness Prediction**

**Rating:** 9
**Confidence:** 4
**Clarity:** The manuscript is well written and ea…

**Strengths:**

+ ProteinGym is comprised of a broad collection of standardized Deep Mutational Scanning (DMS) assays and clinical datasets, covering millions of mutated sequences and encompassing diverse protein families.
+ ProteinGym includes proteins with multiple mutations and indels that most datasets do not.
+ More than 40 high-performing models across zero-shot and supervised settings were implemented and benchmarked.
+ All models are codified with a common interface in the same open-source codebase available in one GitHub repository. New methods can be easily added.
+ The data span different taxa (i.e., humans, other eukaryotes, prokaryotes, and viruses), alignment depths, and mutation types (substitutions vs indels).
+ A website is provided for users to access all benchmarks and get analysis results.


**Additional Feedback:**

N/A

**Correctness:**

The results and description are generally correct. But some statement such as “
the number of possible arrangements for protein sequences as short as 64 residues is larger than the number of atoms in the universe” is not accurate. The minimum length of proteins can be less than 64. The estimated number of proteins in the nature is much less than the number of atoms in the universe.


**Documentation:**

The documentation is fine.

**Ethics:**

There is no ethics concern.

**Limitations:**

In the evaluation of the supervised learning methods, there is no control of the redundancy between the training and test datasets. The protein sequence identity (similarity) between the training and test datasets should be controlled below a user-defined threshold.

**Opportunities For Improvement:**


•	Provide some statistics about data (e.g., # of proteins, # of protein families, labels, mutations, and species).
•	Add some test datasets that have low sequence similarity (e.g., <30% sequence identity) with the training datasets.

•	This benchmark is mostly about prediction of the effects (or fitness) of protein mutations. It is not very clear how the datasets are closely related to typical protein design tasks (e.g., generate protein sequences that may fold into user-defined protein structures). However, the “protein design” is used in the title. Therefore, it may be necessary to adjust the title and some texts to make the topic of this work more specific and clearer.


**Relation To Prior Work:**

The related works in the field are generally well reviewed.

**Summary And Contributions:**

This manuscript presents a large, comprehensive benchmark for predicting the effects/fitness of protein mutations. The datasets including the labels were curated from the Deep Mutational Scanning (DMS) assays and the Clinical data in the field and are much larger than the existing datasets. The work also implemented dozens of zero-shot baseline methods and supervised learning baseline methods and gathered their results, which are valuable for the community to develop new methods. Many different evaluation metrics (Spearman’s correlation and Area Under the ROC Curve (AUC), area under the Precision-Recall Curve, mean square error, and the Matthews Correlation Coefficient (MCC)) are provided for evaluating different prediction tasks according to the nature of the tasks. The results can also be analyzed by the features of the input such as MSA depth, mutational depth, and taxa. The implemented methods are available as open-source programs, which are a valuable resource for both users and developers. The processed datasets used in our various benchmarks are publicly available in a consistent format for training and scoring.

---

> ### Author Response · Authors · 2023-08-17
> **Response to Reviewer vaLL**
>
> We sincerely thank the reviewer for the detailed review and positive feedback. We provide more details on how we integrated your suggestions below.
>
> **C1. Provide some statistics about data (e.g., # of proteins, # of protein families, labels, mutations, and species).**
>
> We have added a new Table 1 in section 3 which summarizes the key attributes of our datasets as suggested.
>
> **C2. Add some test datasets that have low sequence similarity (e.g., <30% sequence identity) with the training datasets.
> In the evaluation of the supervised learning methods, there is no control of the redundancy between the training and test datasets. The protein sequence identity (similarity) between the training and test datasets should be controlled below a user-defined threshold.”**
>
> The mutated sequences in DMS assays or clinical datasets are all very similar within a given protein family (they only differ by a handful of mutated positions such that the similarity between any pair of sequences is beyond 90%), therefore such train-test split is not possible within a given protein family. However, if one is interested in the ability of a (semi-)supervised model to extrapolate to protein families beyond its training set, then the sequence similarity thresholding you are suggesting would be important to consider. We have included a new paragraph in the revision (see edits in red in section 4.2) which discusses this scenario.
>
> **C3. This benchmark is mostly about prediction of the effects (or fitness) of protein mutations. It is not very clear how the datasets are closely related to typical protein design tasks (e.g., generate protein sequences that may fold into user-defined protein structures). However, the “protein design” is used in the title. Therefore, it may be necessary to adjust the title and some texts to make the topic of this work more specific and clearer.**
>
> Thank you for the feedback. We have added a new section in related work that discusses the close connection between these two subfields (see edits in red in the last paragraph of section 2). In short, our perspective is that strong fitness predictors are key components of protein engineering efforts that focus on the design of _functional_ proteins. Based on your feedback, we have also added two metrics (NDCG and Top % recall) that are more adapted to assessing the ability of the various baselines to support protein engineering efforts. We included a discussion of these metrics in section 4.1 (see edits in red) and integrated them with our main results table (Table 2).
>
> **C4. Some statement such as “ the number of possible arrangements for protein sequences as short as 64 residues is larger than the number of atoms in the universe” is not accurate. The minimum length of proteins can be less than 64. The estimated number of proteins in the nature is much less than the number of atoms in the universe.**
>
> Apologies for the slight confusion here. We meant that if one were just considering all theoretically possible amino acid sequences of length 64 -- a subset of all lengths possible -- this would already be much larger than the number of atoms in the universe. We have rephrased this sentence in the revised version.

---

### Official Review · Reviewer_Ckxq · 2023-07-21

**Rating:** 6
**Confidence:** 3
**Clarity:** Overall this paper is well-written an…

**Strengths:**

This paper provides a comprehensive benchmark with different settings, datasets, and a lot of models.

**Additional Feedback:**

No

**Correctness:**

The evaluation methods and experiment design are appropriate and performed correctly.

**Documentation:**

There are sufficient details.

**Limitations:**

The authors have adequately addressed the limitations and potential negative societal impact of their work.

**Opportunities For Improvement:**

1. The paper lacks clarity in introducing detailed information about the datasets, such as the dataset format and data size. This hampers readers' understanding of the task and makes it challenging to replicate experiments efficiently.

2. While the paper focuses on protein sequences, it could benefit from clarifying its relationship to other tasks and methods related to protein structures, such as protein function prediction based on sequences and 3D structures [1][2][3][4], protein structure generation [5][6], and protein inverse folding [7][8]. Understanding these relationships would provide readers with a broader context for the proposed benchmark.

3. Regarding the benchmark's observations, the paper should provide a comprehensive analysis of the results obtained from various methods, assessing their performance and limitations. Additionally, insights into potential future directions and how this benchmark can contribute to the research community would be valuable.

[1] Learning from Protein Structure with Geometric Vector Perceptrons.
[2] Protein Representation Learning by Geometric Structure Pretraining.
[3] Learning Hierarchical Protein Representations via Complete 3D Graph Networks
[4] Continuous-Discrete Convolution for Geometry-Sequence Modeling in Proteins
[5] Broadly applicable and accurate protein design by integrating structure prediction networks and diffusion generative models
[6] Diffusion probabilistic modeling of protein backbones in 3D for the motif-scaffolding problem
[7] Robust deep learning–based protein sequence design using ProteinMPNN
[8] PiFold: Toward effective and efficient protein inverse folding

**Relation To Prior Work:**

It is clearly discussed how this work differs from previous contributions.

**Summary And Contributions:**

This paper provides benchmarks for the task of protein design and fitness prediction. Specifically, it includes 8 benchmarks with 2 mutation types (substitutions and indels), 2 dataset types (DMS assays and clinical datasets), and 2 training settings (zero-shot and supervised). The authors report the performance of over 40 models.

---

> ### Author Response · Authors · 2023-08-17
> **Response to Reviewer Ckxq**
>
> Thank you for the very thoughtful comments and suggestions. We address each of your points of feedback below, including several additions to our initial submission as per your suggestions (Cx: Reviewer’s Concern number x).
>
> **C1: The paper lacks clarity in introducing detailed information about the datasets, such as the dataset format and data size. This hampers readers' understanding of the task and makes it challenging to replicate experiments efficiently.**
>
> We have included a new table (Table 1) in Section 3 which summarizes high-level information about the content of the datasets (more details are present in Tables A2-A3 in appendix).
>
> **C2: While the paper focuses on protein sequences, it could benefit from clarifying its relationship to other tasks and methods related to protein structures, such as protein function prediction based on sequences and 3D structures, protein structure generation, and protein inverse folding. Understanding these relationships would provide readers with a broader context for the proposed benchmark.**
>
> Thank you for the great suggestion. We addressed your feedback as follows:
> 1. We further discuss the relationship with these various methods in section 2.
> 2. Inverse folding models can also be used to predict mutation effects via the perplexity of mutated sequence conditioned on the protein structure. We have integrated two of the most popular inverse folding baselines, ProteinMPNN [1] and ESM-IF1 [2] as a result (now included in Table 2 and Appendix A41 in the revision).
>
> **C3: Regarding the benchmark's observations, the paper should provide a comprehensive analysis of the results obtained from various methods, assessing their performance and limitations. Additionally, insights into potential future directions and how this benchmark can contribute to the research community would be valuable.**
>
> Based on your suggestion, we have significantly expanded the discussion of our results in section 5, including the new baselines and metrics integrated as part of this rebuttal (see modified text in red, new Figure 2B in section 5 and Figure A1 in Appendix).
>
> **References**
>
> [1] Dauparas, Justas et al. “Robust deep learning based protein sequence design using ProteinMPNN.” Science, 2022.
> [2] Hsu, Chloe et al. “Learning inverse folding from millions of predicted structures.” ICML, 2022.

---

> > ### Comment · Reviewer_Ckxq · 2023-08-29
> >
> > I appreciate the authors' response and have increased my score.
> >
> > Could the authors further summarize this paper (preferably using bullet points and short answers):
> > 1. About the task: Why this task is important and challenging? What is the key difference with other tasks (tasks used in the reference papers I mentioned earlier)? ...
> >
> > 2. About the results: What are the key observations from the results? Any insights for future research? ...
> >
> > 3. Other points (contribution, novelty, impact, etc.) the authors may want to summarize.

---

> > > ### Author Response · Authors · 2023-08-29
> > > **Response to Reviewer Ckxq (Part 1)**
> > >
> > > Dear reviewer,
> > >
> > > Thank you for your response and the positive feedback. We address the few questions you raised below
> > >
> > > **About the task: Why is this task important and challenging? What is the key difference with other tasks (tasks used in the reference papers I mentioned earlier)?**.
> > > This work introduces the most extensive set of benchmarks to assess fitness prediction models. Fitness prediction aims at understanding the relationship between the representation of a protein (sequence or structure) and whether it is functional. Fitness prediction is critical to many applications, namely:
> > > 1. **Protein design [1,2]**:
> > > - **Importance**: ability to design new sequences that are functional, to be used as therapeutics, new materials, plastic degraders, and many more.
> > > - **Challenge**: the potential design space is massive, and functional protein sequences are sparse, which is further exacerbated by the fact that functionality is tightly dependent on the environment (e.g., temperature, pH, host organism)
> > > 2. **Clinical variant effect prediction [3,4]:**
> > > - **Importance**: ability to predict whether certain genetic mutations will translate into functional proteins and, in turn, leading to pathologies downstream
> > > - **Challenge**: clinical labels are sparsely available and scattered across many unrelated genes/proteins leading to issues during training (overfitting) and evaluation (data leakage resulting from careless cross-validation schemes that ignore dependencies between mutations happening at the same positions)
> > > 3. **Viral escape [5,6]**:
> > > - **Importance**: ability to predict which mutations in viruses will maintain functionality and escape human immunity. Inform pandemic preparedness and vaccine design efforts
> > > - **Challenge**: labels are sparse, especially early in a pandemic
> > >
> > > The papers you had referenced either focus on protein structure prediction, protein representation learning or inverse folding:
> > > - **Inverse folding** models are typically used in protein engineering to generate sequences that are likely to fold in a desired input 3D structure. Even with the latest papers you cited (e.g., RFdiffusion) the large majority of these sequences will not be functional in practice (less than 20% in the de novo binder design experiments from Watson et al. [7]) and are typically used as starting precursor for subsequent protein engineering efforts that will turn them into functional proteins, using fitness models such as the ones covered in our work or pure experimental method (eg., directed evolution)
> > > - **General-purpose protein representation** is helpful in practice when leveraged for a specific task. For instance some of the learned embeddings could be used as input to some of the baselines we discussed in the supervised setting. They are however not useful in the zero-shot setting
> > > - **Protein structure prediction** tools aim at predicting the 3D structure of a protein given the corresponding amino acid sequence. These tools can be used in many settings -- we actually used AlphaFold2 to obtain the structures that are used to condition ProteinMPNN and ESM-IF1 in the rebuttal addressing your earlier feedback
> > >
> > > **References**
> > > [1] Shin, Jung-Eun et al. “Protein design and variant prediction using autoregressive generative models.” Nature Communications (2019).
> > > [2] Nijkamp, Erik et al. “ProGen2: Exploring the Boundaries of Protein Language Models.” ArXiv abs/2206.13517 (2022).
> > > [3] Frazer, Jonathan et al. “Disease variant prediction with deep generative models of evolutionary data.” Nature, 2021.
> > > [4] Brandes, Nadav et al. “Genome-wide prediction of disease variant effects with a deep protein language model.” Nature genetics (2023).
> > > [5] Hie, Brian L. et al. “Learning the language of viral evolution and escape.” Science (2020).
> > > [6] Thadani, Nicole N. et al. “Learning from pre-pandemic data to forecast viral escape.” bioRxiv (2023).
> > > [7] Watson, Joseph L. et al. “De novo design of protein structure and function with RFdiffusion.” Nature (2023)

---

> > > > ### Author Response · Authors · 2023-08-29
> > > > **Response to Reviewer Ckxq (Part 2)**
> > > >
> > > > **About the results: What are the key observations from the results? Any insights for future research?**
> > > >
> > > > - On our DMS benchmarks, the relative performance of various baselines exhibits strong variance across assays -- robust assessment requires scale, and we do that at an unprecedented scale in terms of both number of assays and mutants
> > > > - We observe that pure protein language models -- even the most recent ones (eg., ESM2, Progen2) are not competitive with MSA-based baselines for fitness prediction. The best baselines are either methods trained purely on MSAs, or hybrid methods that combine aspects of protein language models with MSA / homology (eg., Tranception/TranceptEVE, MSA Transformer). This is consistent with preliminary findings in our Tranception paper -- although we had only a handful of baselines at the time (9 then vs 40 now)
> > > > - Inverse folding baselines are not performing well on our DMS benchmark - meaning they generate sequences that are likely to adopt a certain fold, but since fitness prediction ability is limited, many of the de novo generated sequences will not be functional. This opens the door for exciting development in the field by combining the relative strengths of inverse folding and latest techniques for fitness modeling
> > > > - Comparing results on the supervised vs zero shot regimes helped identify model architectures that are more helpful for zero-shot fitness prediction (ie., autoregressive architectures) and others that provide superior embeddings to be used in the supervised settings (e.g., MLM baselines)
> > > > - Clinical benchmarks helped surface the fact that models trained in a fully unsupervised way can outperform supervised baselines to predict the effects of genetic mutations, consistent with findings in EVE [Frazer et al], and that superior zero-shot fitness predictors tend to translate into superior performance on clinical benchmarks (for instance ESM-1b < EVE < TranceptEVE consistently on both benchmarks)
> > > >
> > > > **Other points (contribution, novelty, impact, etc.) the authors may want to summarize.**
> > > > - Besides its scale and the extensive list of baselines included, ProteinGym is a collection of complementary benchmarks, and it is by combining different synergistic datasets, metrics and training modes together that new insights emerge. For a more detailed discussion on this, please see our last two comments to Reviewer 3bqN (Clarifications regarding novelty - Part 1 and 2) where we review our key contributions and assess them in terms of novelty, importance, and relevance to the NeurIPS community.
> > > > - Beyond benchmarking capabilities, the curated data we built as part of this work will serve as foundation for the development of the next generation of protein models that make use of the 3.7M labels we put together across assays and clinical datasets

---

> > > > > ### Comment · Reviewer_Ckxq · 2023-08-29
> > > > >
> > > > > Thanks for the authors' response. My concerns are addressed.

---

### Official Review · Reviewer_3bqN · 2023-07-21
**Limited novelty**

**Rating:** 2
**Confidence:** 4
**Correctness:** Yes
**Clarity:** Yes

**Strengths:**

The paper is clearly written and presents a well-designed benchmark.

**Additional Feedback:**

What fundamental innovation is being proposed in this work? How does that differ from the authors' previous work while also aligning with the goals of the NeurIPS dataset track?

**Documentation:**

Yes

**Limitations:**

Yes

**Opportunities For Improvement:**

**Double dipping**: The authors have already published the ProteinGym benchmark (Notin et al. 2022) with the same name at ICML last year. In fact, both the public GitHub repository linked to this submission and the new repository included in the supplementary information ask users to cite the previous publication (from ICML 2022) when they use ProteinGym. Moreover, the new repository attached to this submission appears to share much of the same code with the public version. So, clearly the idea for the benchmark and much of its implementation was already linked to a previous publication.

**Not only are the contributions similar but so is the text from previous published paper**. Some examples below:

**Notin et al 2022**: To build this benchmark we therefore prioritized assays where both the experimentally-measured property for each mutated protein is expected to reflect the role of the protein in organism fitness …
**Neurips 2023 submission**: (Line 140) In developing ProteinGym, we prioritized assays where the experimentally measured property for each mutant protein is likely to represent the role of the protein in organismal fitness

**Notin et al 2022**: … spans a diverse protein families (eg., kinases, ion channel proteins, g-protein coupled receptors, polymerases, transcription factors, tumor suppressors)
**Neurips 2023 submission**: (Line 145) It encompasses diverse protein families, such as kinases, ion channel proteins, G-protein coupled receptors, polymerases, transcription factors, and tumor suppressors.

**Notin et al 2022**: The relationship between protein function and organism fitness has been shown to often be non-linear (Boucher et al., 2016).
**Neurips 2023 submission**: (Line 179) Previous studies have demonstrated that the link between protein function and organism fitness often exhibits a non-linear nature Boucher et al.

**Notin et al 2022**:  As such, we use Spearman’s rank correlation coefficient between model scores and the experimental mea-surements as the standard measure of model performance (Riesselman et al., 2018;
**Neurips 2023 submission**: (Line 180) Therefore, we use Spearman’s rank correlation coefficient as a primary metric for model performance, comparing model scores with experimental measurements Riesselman et al.

**Incremental improvement over past work**: Even if these serious issues with novelty and originality are overlooked, this paper presents only an incremental improvement in the size of an already existing, published benchmark. The authors thus have a greater burden to describe the novelty in this work in comparison to their previous work. Notin et al. 2022 already includes substitution benchmarks and an indel benchmark, albeit with fewer baseline models and assays. For instance, 8 out of the 17 rows in Table 1 were already published _as is_ in Notin et al 2022. And the new rows only correspond to new published models that have been applied to the same benchmark.


**Relation To Prior Work:**

No

**Summary And Contributions:**

The authors present a benchmark for standardizing the evaluation of protein fitness prediction across multiple experimental assays.

---

> ### Author Response · Authors · 2023-08-17
> **Response to Reviewer 3bqN**
>
> We thank the reviewer for the time spent on reviewing our work. We think there are substantial misunderstandings about what is new in our work since the last publication [1]. This may have arisen from the fact we continuously update the public GitHub repo with newer models and data since that publication.
> We first make important clarifications relating to these misunderstandings, and then discuss how we addressed the other points of feedback in the revision.
>
> **Important clarifications**
>
> 1. _“Double-dipping”_: we very clearly state in the “Related work” section the fact that we build on top of a prior version of ProteinGym (see lines 101-105), and that we have made significant changes on top of it as discussed in the subsequent sections 3 and 4 of the manuscript. We refer the reviewer to point A of the response to all reviewers for an in-depth discussion about how our submission differs from the version of the benchmark included in [1] (this detailed comparison is now included in the revision, in Table A1).
>
> 2. Relatedly, the reviewer indicates that we do not discuss the relation to prior work sufficiently (“Relation to prior work: No”). However, we do think prior art was extensively covered (see lines 61-105 in our submission). We kindly ask the reviewer to state which relevant prior works have not been discussed otherwise.
>
> 3. _“In fact, both the public GitHub repository linked to this submission and the new repository included in the supplementary information [..] the new repository attached to this submission appears to share much of the same code with the public version. So, clearly the idea for the benchmark and much of its implementation was already linked to a previous publication”_.
> There seems to be a confusion between what is published work with work that is publicly available. The code that corresponds to prior published work [1] is [this checkpoint](https://github.com/OATML-Markslab/Tranception/tree/be4bb465b20e1bfd5b8f967fbd7a8f657d57d3f6) from the _Tranception_ GitHub repository (released right before ICML 2022).
> We have released several new baselines in the public repository of [ProteinGym](https://github.com/OATML-Markslab/ProteinGym) since, but all these changes were done _after_ [1] and are not published anywhere. Lastly, while the README file of this public repo and the one from the submission are similar, their actual contents are substantially different (again, we refer to point A of response of all reviewers, in particular point #A5).
>
> 4. _“For instance, 8 out of the 17 rows in Table 1 were already published as is in Notin et al 2022. And the new rows only correspond to new published models that have been applied to the same benchmark”_.
> First of all, there are 40 (not 17) baselines total on that benchmark (methods in that table are a subset of all baselines implemented and available in Table A4). Second of all, none of these results have been published anywhere and they were all computed by ourselves. This often involved custom code development to adapt public codebases that were either not setup for fitness prediction, or would not handle all required edge cases (e.g., sequences longer than the standard context length of protein LMs, scoring multiple mutants).
>
> **Other comments**
>
> **C1. Not only are the contributions similar but so is the text from previous published paper**.
> As discussed above, we see our contributions as being radically different than [1]. We mirrored some elements of language of ProteinGym v0.1, as there are general rules about how we constructed certain aspects of our benchmarks that are still applicable to the newer version of ProteinGym. We have slightly rephrased some of these sentences in the revised version, but we firmly believe that the general content of these sentences is still relevant despite the benchmarks being significantly different.

---

> > ### Author Response · Authors · 2023-08-17
> > **Response to Reviewer 3bqN (continued)**
> >
> > **C2. Even if these serious issues with novelty and originality are overlooked, this paper presents only an incremental improvement in the size of an already existing, published benchmark.**
> >
> > As discussed in point A to all reviewers, there are substantial contributions of this submission over previously published work: entirely new datasets (we insist on the clinical benchmarks in particular), new training regimes (eg., supervised setting), 5x more baselines (zero-shot and supervised), a radically enhanced codebase that now includes the code to score the main baselines, and a dedicated website. Since the large majority of these points were entirely absent from their review, it was not clear whether the reviewer did not think these were meaningful or if they were overlooked.
> >
> > Even though we strongly believe the aforementioned contributions can hardly be characterized as incremental, we further reinforce our contributions in this rebuttal by adding 157 new assays to our DMS benchmarks, effectively doubling the number of proteins and mutants compared to DMS assays in [1]. We also added new baselines and performance metrics in response to the feedback from other reviewers.
> >
> > **References**
> >
> > [1] Notin, Pascal et al. “Tranception: protein fitness prediction with autoregressive transformers and inference-time retrieval.” ICML, 2022.

---

> > > ### Comment · Reviewer_3bqN · 2023-08-21
> > > **Limited novelty stands**
> > >
> > > **Limited novelty**: Given this set of claims (A1-6 and B), I still remain unconvinced on the originality and novelty of this submission, both of which are key criteria listed for judging datasets as per the reviewer guidelines. I believe that only A3 adds meaningful novelty to the benchmark that is significant but not enough to carry the paper. From a novelty perspective, I don't think it is a significant change to switch model training from the zero-shot setting to a supervised setting (A4), nor is it significant to incorporate a few more existing fitness prediction models (A5, B). The rest of the contributions are either new datasets or two additional metrics that are not novel.
> > >
> > > To reiterate from my original comment, this does not take away from the importance and usefulness of this benchmark overall. The authors deserve credit for that, and have rightly received (most of) it already through their ICML publication.

---

> > > > ### Author Response · Authors · 2023-08-24
> > > > **Clarifications regarding novelty (Part 1)**
> > > >
> > > > We first point out that “lack of novelty” over one’s prior work is synonymous with “double dipping” and that, right after retracting that criticism, the reviewer appears to be rebranding it under a different name.
> > > > For the purpose of having a constructive discussion with the reviewer, we address these comments regardless. We sought to be as fact-based and detailed as possible in the below, pointing out the several ways in which we found the criticism to be invalid or missing the broader picture. We also insist on the impact and relevance to the NeurIPS community of these contributions, which are also important reviewing criteria for new datasets and benchmarks at NeurIPS.
> > > >
> > > > **1. New DMS benchmarks.** The reviewer describes our contributions here as being just “new datasets”.
> > > > - **Novelty**: this may be stating the obvious, but introducing “new datasets” is one of the stated objectives of the “NeurIPS datasets and benchmarks” track, and “_new_ datasets” are _novel_, by definition.
> > > > - **Impact**: there are now more than twice as many DMS benchmarks in v1.0 than there were in v0.1 (257 vs 94 assays total). That is, to build the current version, we curated 163 novel DMS assays (105 substitutions, 58 indels). To put things in perspective, when developing ProteinGym v0.1, we had curated 56 new assays (49 subs, 7 indels) on top of the prior benchmark (DeepSequence). We believe this is a significant contribution that is still unaccounted for in the review. Scale and diversity strengthen the robustness of the evaluations in the benchmarks, which is critical to build reliable fitness prediction benchmarks as abundantly discussed in our paper already (abstract, introduction, etc).
> > > > - **Relevance to NeurIPS community**: as the reviewer acknowledged, the feedback on the ProteinGym v0.1 and subsequent public extensions has been overwhelmingly positive. These have been used for benchmarking purposes, but not only. With the release of ProteinGym v1.0, we have curated for the community an additional ~1.7 million experimental labels that can be used for training standalone models, fine tuning large protein language models, meta learning across tasks, and many more.
> > > >
> > > > **2. New clinical benchmarks**
> > > > - **Novelty**: we appreciate that the reviewer now characterizes our contributions here as adding “meaningful novelty to the benchmark”. We also wanted to highlight the fact that what is really powerful here is the combination of clinical datasets and deep mutational scanning assays together within the _same_ benchmark. For instance, there is a known issue of “circularity” with clinical labels: certain supervised models trained on clinical labels have been used as supporting evidence by experts (e.g., as part of broader annotation frameworks such as ACMG criteria) to label certain variants. Assessing their performance on the resulting set of labels therefore tends to overestimate their performance, an issue which can be detected by leveraging an independent source of evidence such as DMS assays. It is for that reason that Fig2A plots the performance of the various baselines on these two datasets simultaneously (the x-axis corresponds to the clinical dataset performance, and the y-axis to the DMS performance). So one should not just be looking at the individual value coming from one additional data source in isolation from the rest, but also at the synergies created when that new data source is being combined with other elements of the benchmark.
> > > > - **Impact**: What is at stake here is helping to drive forward the next generation of models that predict the effects of genetic mutations in humans. This has critical potential in clinical and therapeutic applications.
> > > > - **Relevance to NeurIPS community**: A lot of thought and work has been put in to address non-trivial issues inherent to these datasets, such as mapping issues (for substitutions and indels) and creating controls (for indels) (please refer to our appendix for more details). We believe these resources will be immensely useful for peers from the ML community that may not have the biostatistics know-how to curate such datasets. This is aligned with the key objectives of the NeurIPS datasets and benchmark track, as stated in the [Call for datasets & benchmarks](https://nips.cc/Conferences/2023/CallForDatasetsBenchmarks), as well as  the [initial blog post](https://neuripsconf.medium.com/announcing-the-neurips-2021-datasets-and-benchmarks-track-644e27c1e66c) that discusses why this track was created in the first place.

---

> > > > > ### Author Response · Authors · 2023-08-24
> > > > > **Clarifications regarding novelty (Part 2)**
> > > > >
> > > > > **3. New baselines and metrics**
> > > > > - **Novelty**: Describing our contribution with respect to metrics as “just two additional metrics that are not novel” is missing the broader point. The addition of these 2 metrics, which are geared towards protein design unlike other metrics in v0.1,  _critically_ builds a bridge between fitness prediction and protein design, two important subdomains of computational biology that are often perceived as separate. It is this connection that is novel here, not the metrics themselves. Building this bridge has always been a primary objective of this work, as evidenced by our choice of title for the paper (see also the additional paragraph we added at the end of the background section during revisions). For the same reason, we had added many baselines from the two subfields (just to give two examples out of 40 — ProGen for protein design, GEMME for fitness prediction). Characterizing our contribution as “incorporating a few more existing fitness prediction models” fails to appreciate what is really at stake here. Bringing together these models from different subfields in the same unified benchmark helps build new connections that we have not seen in prior benchmarks, including ProteinGym v0.1.
> > > > > - **Importance**: Furthermore, the addition of inverse folding baselines builds another critical connection between structure-based methods for de novo design and sequence-based models for fitness prediction or design. Over the past few years, we have seen countless inverse folding papers assessing their performance using relatively limited metrics such as “recovery rate”, simply because the prior inverse folding benchmarks they had to compare against were all using that assessment. By making the connection with fitness prediction more explicit, we hope that the DMS benchmarks included in this paper will be used as complementary evaluations to inform the development of the next generation of inverse folding models.
> > > > > - **Relevance to NeurIPS community**: To clarify, we do not claim that we are the first to appreciate the aforementioned connections, but we find them to be relatively poorly understood by the broader ML for computational biology community and there are no existing benchmarks that facilitate these connections or allow straightforward comparison of these models side by side. Additionally, the fact that we are open sourcing the main baselines will be a valuable resource to the community. This includes critical extensions of otherwise open sourced baselines (eg., ESM), as well as baselines that are open sourced in full for the very first time (eg., TranceptEVE). Lastly, developing such a benchmark is not trivial -- both from a conceptual and practical standpoint. One of these practical challenges is that as one doubles the number of assays, doubles the number of training regimes, doubles the number of dataset types (DMS vs. clinical) and multiplies by 5 the number of baselines, the corresponding compute budget increases dramatically. As a conservative estimate, ProteinGym v1.0 involved over **200k GPU hours** for its development. In contrast, ProteinGym v0.1 involved about **5k GPU hours**.
> > > > >
> > > > > In the same spirit as point #3, incorporating a supervised training regime within the same benchmark as a zero-shot regime helps uncover new insights that would be missed by looking at these in isolation. There have been several excellent protein benchmarks focusing on the supervised setting alone as discussed in the related work section (e.g., FLIP). Besides the fact that we enable the assessment of supervised baselines at an unprecedented scale, additional novelty comes from bringing the supervised and zero-shot regime side-by-side which, to our knowledge, has never been done before. This has led to interesting insights, such as those discussed in section 5 regarding the respective strengths of autoregressive vs. MLM depending on the training regime.
> > > > >
> > > > > To conclude, by looking at the merits of our various contributions in isolation from one another, one may be missing the bigger picture. One of the main strengths of a holistic set of benchmarks like ProteinGym v1.0 -- unlike v0.1 -- is that it leads to non-obvious insights by bringing seemingly separate subfields and components together. It is for that reason that ProteinGym v1.0 is _fundamentally_ different in _nature_ compared to v0.1, and it is also for that reason that is should not perceived as a mere extension of an existing benchmark “with just more assays and baselines” of the same kind.

---

> > ### Comment · Reviewer_3bqN · 2023-08-21
> > **Text is confusing on claims**
> >
> > 2. (and 1) I thank the authors for elaborating on the differences between this work and their previous publication. This was exactly the ‘relationship to prior work’ that I believe was missing in their original submission and in fact is still missing in the current version. Lines 101-105 that they quote above do not provide any of this detail.
> >
> > 3. Thanks also for clarifying which checkpoint is relevant for comparing codebases. I don’t think it is obvious that the 27th commit in their public repository that cites Notin et al. 2022 is the point of reference we should be using for comparison.
> >
> > **Submission text now does not reflect true claims**: While these clarifications above (and not the paper itself) have made clear that this work is indeed not double dipping, it has also clearly limited the scope of their core contribution to additional datasets, a few metrics, a new training mode and some more benchmark models. I don't think the website is a significant contribution.
> >
> > However, to be honest about their claims, the authors should also reflect these changes in their manuscript. For instance, I would change all instances of the phrase ‘introduce ProteinGym’ to ‘extend ProteinGym’. For instance, the abstract could be rewritten as follows:
> >
> > >Despite an increase in machine learning-based protein modeling methods, assessing their effectiveness is problematic due to the use of distinct, often contrived, experimental datasets and variable performance across different protein families. Addressing these challenges requires scale. **While existing protein fitness prediction benchmarks such as ProteinGym have provided a mechanism for standardizing this evaluation we believe it to be insufficient. Here we _extend_ ProteinGym with** a broad collection of standardized deep mutational scanning assays, spanning millions of mutated sequences and covering protein families from diverse taxa, functions and depth of homologous sequences.
> >
> > The novel contribution of this paper should only be judged on the basis of the extension that is described in bullets A1-6 and B of the author response and not the paper text which is confusingly written. For example, lines 43-55 in this submission that are meant to summarize their core contribution could have been directly used also in Notin et al. 2022, if you just remove a few terms like 'clinical datasets', 'multiple mutants' and 'supervised setting'. In my opinion, a truer summarization of their contribution would be more like:
> >
> > > Here, we present an extension of ProteinGym that adds 31 new sequence-only zero-shot baselines and 17 new indel baselines. Clinical substitution and indel benchmarks were incorporated, offering a broader perspective on proteins through annotations from thousands of genes and expert reviews. Supervised benchmarks and two additional metrics were also added. We have open sourced our code and added a dedicated website to enhance accessibility.
> >
> > For further constructive discussion, it would be helpful if the current manuscript more precisely reflected its true claims.

---

> > > ### Author Response · Authors · 2023-08-24
> > > **Clarifications regarding claims**
> > >
> > > We thank the reviewer for the additional responses, and for retracting the main points of criticism from their initial review — the “double dipping” point in particular. We are glad to see the clarifications in our last response were helpful to them in reaching that conclusion. We would like to stress that the contributions A1-A5 were all fully discussed in the original submission — some of these points being subsection titles or paragraph titles — we only summarized them in our rebuttal. These points were however entirely absent from the reviewer’s description of the strengths of the paper (_“The paper is clearly written and presents a well-designed benchmark”_). For instance, the addition of the clinical benchmarks was not mentioned, while the reviewer now concedes that it “adds meaningful novelty to the benchmark that is significant” (we address the concerns about the “lack of novelty” in response to your other comment below).
> > >
> > > Given these important concessions, it would be highly appreciated if the reviewer could adjust their initial review accordingly — in particular: 1) removing claims that are now retracted (double dipping and lack of discussion of prior literature), 2) describing strengths in a way that is in line with their current appreciation of the contributions of this paper, as well as 3) updating their score in a way that is commensurate with their revised assessment given the major retractions.
> > >
> > > As for the suggested text edits, we are open to adjust the language further in order to alleviate any residual concerns on the clarity of our contributions, and appreciate the opportunity to have a constructive discussion with the reviewer on that point. We made these text edits in blue in the newly uploaded revision, focusing on abstract and introduction as suggested. As we disagree on the nature and significance of our contributions (please see our response to your next comment), the edits we made are different from the ones suggested. We sought to ensure that all sentences in the abstract or introduction would uniquely characterize ProteinGym v1.0, and would not be applicable to ProteinGym v0.1 (e.g., design was not a focus of that benchmark, it did not include clinical datasets, metrics & baselines were limited to mutation effects, codebase was not open sourced). Please let us know whether you agree with the updated language.

---

### Official Review · Reviewer_dron · 2023-07-22
**Review of ProteinGym: Large-Scale Benchmarks for Protein Fitness Prediction and Design**

**Rating:** 7
**Confidence:** 3
**Correctness:** I didn’t find errors from the paper.
**Clarity:** This paper is well-written.

**Strengths:**

- Covers a wide range of data, mutation types
- Provides a detailed description of the datasets, evaluation metrics, and baseline models used in the benchmarks
- Reports the performance of over 40 high-performing models across zero-shot and supervised settings, thus providing insights for future researchers.
- All models are codified with a common interface in the same open-source codebase, promoting consistency and ease of use, enhancing transparency and accessibility in the prediction of protein fitness landscapes


**Additional Feedback:**

None

**Documentation:**

There is reasonable information on the website and the repo's README.

**Limitations:**

See above

**Opportunities For Improvement:**

This study could further gain from the integration of a wider selection of models and search algorithms. Also, incorporating metrics other than Spearman and AUC could provide a more thorough assessment of the models.

**Relation To Prior Work:**

The relation to prior works is generally clearly claimed.

**Summary And Contributions:**

The focus of this paper is on the process of utilizing standard testing models for predicting the impacts of mutations, a significant problem in the field of biology. The proposed benchmark, named ProteinGym, provides a comprehensive reference scale for evaluating the effectiveness of protein modeling techniques based on machine learning. The benchmarks include a range of datasets with various mutation types, a broad collection of models and evaluation metrics. The authors provide detailed explanations of the datasets, evaluation metrics, and initial models used in the benchmarks. Essentially, ProteinGym proves to be a beneficial tool for academics and professionals involved in protein design and predicting fitness.

---

> ### Author Response · Authors · 2023-08-17
> **Response to Reviewer dron**
>
> We thank the reviewer for the time reviewing our submission, the positive feedback and the suggestions to further strengthen our work. We discuss how we integrated these suggestions in the revision below.
>
> **C1: This study could further gain from the integration of a wider selection of models and search algorithms**
>
> We have integrated two additional baselines (ProteinMPNN [1] and ESM-IF1 [2]) in the revision. Please let us know if any other open source baselines would be critical to include, and we will do our best to integrate them by the end of the discussion window.
>
> **C2: incorporating metrics other than Spearman and AUC could provide a more thorough assessment of the models.**
>
> We added two new metrics (NDCG and Top 10% recall) that are geared towards assessing the usefulness of models for protein engineering efforts (see Table 2 in revision). Besides AUC and Spearman, our original submission also contained Matthew’s Correlation Coefficient (MCC) evaluations (line 201) but the corresponding results were buried in the github repository, and we have made them more prominent in Table 2.
>
> **References**
>
> [1] Dauparas, Justas et al. “Robust deep learning based protein sequence design using ProteinMPNN.” Science, 2022.
> [2] Hsu, Chloe et al. “Learning inverse folding from millions of predicted structures.” ICML, 2022.

---

### Official Review · Reviewer_5XQY · 2023-07-23

**Rating:** 6
**Confidence:** 2
**Correctness:** N/A
**Clarity:** N/A

**Strengths:**

- Clear logic and writing.
- The realm and scope of this paper are explicitly explained.
- The benchmarks are comprehensive.

**Additional Feedback:**

N/A

**Documentation:**

N/A

**Limitations:**

The only limitation of this work is that it mainly focuses on the protein sequences, while the protein 3D structures can reveal more important and interesting insights. However, this is definitely beyond the scope of this paper, and it can be a promising future direction.

Few tools for benchmarking 3D proteins are coming out recently. Authors can easily find them and merge them into ProteinGym. (I cannot comment on more details due to a potential conflict)

**Opportunities For Improvement:**

- One thing I find confusing is, what is the relation between this ProteinGym paper and the ProteinGym in the Tranception paper? Or what are the overlaps between them? Maybe authors can help explain this.

**Relation To Prior Work:**

It has been sufficiently discussed in Sec 2.

**Summary And Contributions:**

The authors propose ProteinGym, a tool designed for protein fitness prediction and equipped with structured evaluation. It also includes 40 models across zero-shot and supervised settings.

---

> ### Author Response · Authors · 2023-08-17
> **Response to Reviewer 5XQY**
>
> We thank the reviewer for their thoughtful review and suggestions for improvements.
> We address the various points raised below (Cx: Reviewer’s Concern number x).
>
> **C1: One thing I find confusing is, what is the relation between this ProteinGym paper and the ProteinGym in the Tranception paper? Or what are the overlaps between them? Maybe authors can help explain this.**
>
> We kindly refer the reviewer to point A in the response to all reviewers. We have now integrated this detailed description in the revision (Table A1 in appendix).
>
> **C2: The only limitation of this work is that it mainly focuses on the protein sequences, while the protein 3D structures can reveal more important and interesting insights. However, this is definitely beyond the scope of this paper, and it can be a promising future direction. Few tools for benchmarking 3D proteins are coming out recently. Authors can easily find them and merge them into ProteinGym.**
>
> We thank the reviewer for the great suggestion and have integrated two of the most popular structure-based inverse folding methods (ProteinMPNN [1] and ESM-IF1 [2]) in the revision (see Table 2 and Appendix section A.4.1)
>
> **References**
>
> [1] Dauparas, Justas et al. “Robust deep learning based protein sequence design using ProteinMPNN.” Science, 2022.
> [2] Hsu, Chloe et al. “Learning inverse folding from millions of predicted structures.” ICML, 2022.

---

### Author Response · Authors · 2023-08-17
**Response to all reviewers**

Dear reviewers,

We sincerely thank you for the time spent engaging with our paper and really appreciate the thoughtful comments. Based on your feedback, we have further enriched the benchmarks with additional DMS assays, evaluation metrics and baselines, and have clarified all points you had raised. We believe the submission is much stronger as a result. We summarize the key points of feedback and how we addressed them as follows:

A. **Differences with the first version of ProteinGym published in the Tranception paper [1] (Rev. 3bqN and 5XQY)**

We want to underscore that, while we decided to keep using the ProteinGym name for this new version of the benchmarks, we have made _substantial_ improvements on many fronts since [1]. We refer the reviewers to Table A1 in the revised version, which summarizes how ProteinGym relates to and builds on prior published benchmarks. To disambiguate, we versioned the ProteinGym benchmarks (and will continue to do so as we continue to improve them):
- The original ProteinGym version in the Tranception paper is v0.1
- The current version, which incorporates all feedback from reviews, is v1.0

About the specific contributions included in 1.0 that were absent from 0.1:

A1. We implemented 31 new sequence-only zero-shot baselines, and provide the corresponding assay-level performance, as well as all mutant-level predictions:
- Substitutions: 40 baselines in v1.0 vs 9 in v0.1
- Indels: 20 baselines in v1.0 vs 3 in v0.1

A2. We developed a dedicated website to facilitate access and analysis of results (did not exist when v0.1 was published)

A3. We added clinical substitution and indel benchmarks. They provide a complementary viewpoint to DMS assays, as they cover a much wider set of proteins, with high-quality annotations from by human experts.
- We built the substitution dataset based on ClinVar annotations available for ~3.7k genes, that we processed and mapped to protein sequences. Several baselines were obtained from the dbNSFP [2], to which we added ESM-1b scores from Brandes et al. [3], and computed EVE [4] and TranceptEVE [5] scores.
- We built the indel dataset based on annotations from ClinVar for ~1.6k genes and gnomAD common variants as pseudo-controls (see Sec. 3.2 and appendix A.3.2 for details); we implemented 12 baselines on this benchmark (Appendix table A8)

A4. We added supervised benchmarks in addition to zero-shot benchmarks, for substitutions and indels (v0.1 was focusing on the zero-shot setting only)

A5. We are open sourcing the code we developed for the main baselines and processing raw datasets (not done in v0.1, nor in the public repo yet). This includes the code for baselines that have not yet been publicly released, such as TranceptEVE, as well as extensions for several publicly-available baselines (e.g., extending them all to handle fitness prediction in ProteinGym, extend ESM baselines to score sequences beyond context-length, extend VESPA [6] to score multiple mutants).

A6. We added new DMS assays, performance metrics and structure-based baselines based on feedback from reviewers (more on this in point B below).

All points above are, in our view, important contributions which will be valuable resources to the community. As further clarification, points A1 & A2 were publicly released _after_ v0.1 but never published (except ESM2 [7] baselines that have not been made public yet); points A3, A4, A5 are major contributions not yet public; and changes in point A6 correspond, for the most part, to additional improvements done for this rebuttal, which we discuss in the next paragraph in more detail.

B. **Main additions to ProteinGym based on reviews**

Based on the feedback from all reviewers we made the following enhancements:
- Added 92 new substitution DMS assays and 65 new indel DMS assays [Based on feedback from Rev. **3bqN**; this is on top of the 13 substitution assays and 2 indel assays that were in our submission already but not present in [1]]
- Added 2 new performance metrics -- NDCG and Top 10% recall -- for the DMS zero-shot benchmarks, which are focused on protein design objectives (see new paragraph in section 4.1) [Based on feedback from **Ckxq**, **vaLL** and **dron**]
- Added 2 additional structure-based models (ProteinMPNN [8] and ESM-IF1 [9]) in the set of DMS baselines for zero-shot substitution effects prediction. We also provide the predicted structures used along the way as additional assets for each DMS assay [Based on feedback from **5XQY**, **dron** and **Ckxq**]

---

> ### Author Response · Authors · 2023-08-17
> **Response to all reviewers (continued)**
>
> We updated the submission based on the above:
> - All new DMS assays have been integrated to Tables A2, A1 & A13
> - New performance metrics and structure-based baselines have been added to Table 2
>
> Note that scoring all baselines on the new 157 DMS assays, both in the zero-shot and supervised settings, for substitutions and indels, represents a substantial compute workload that is still in progress as we write this response. We will update Table 1-3 and A4-A11 as soon as completed.
>
> **References**
>
> [1] Notin, Pascal et al. “Tranception: protein fitness prediction with autoregressive transformers and inference-time retrieval.” ICML, 2022.
> [2] Liu, Xiaoming et al. “dbNSFP v4: a comprehensive database of transcript-specific functional predictions and annotations for human nonsynonymous and splice-site SNVs.” Genome Medicine,  2020.
> [3] Brandes, Nadav et al. “Genome-wide prediction of disease variants with a deep protein language model.” bioRxiv, 2022.
> [4] Frazer, Jonathan et al. “Disease variant prediction with deep generative models of evolutionary data.” Nature, 2021.
> [5] Notin, Pascal et al. “TranceptEVE: Combining Family-specific and Family-agnostic Models of Protein Sequences for Improved Fitness Prediction.” NeurIPS, LMRL workshop, 2022.
> [6] Marquet, Céline et al. “Embeddings from protein language models predict conservation and variant effects.” Human Genetics, 2021.
> [7] Lin, Zeming et al. “Evolutionary-scale prediction of atomic level protein structure with a language model.” Science, 2023.
> [8] Dauparas, Justas et al. “Robust deep learning based protein sequence design using ProteinMPNN.” Science, 2022.
> [9] Hsu, Chloe et al. “Learning inverse folding from millions of predicted structures.” ICML, 2022.

---

### Comment · Area_Chair_KGHA · 2023-08-17
**Please respond to rebuttals!**

Reviewers,

Please try to respond to rebuttals early to give the authors as much opportunity as possible to improve their paper. This is especially important if you gave the paper a marginal or borderline score.

Also, my opinion is that the paper is different enough from the Transception paper to not be double-dipping. Please focus further discussion on the content of the paper.

---

### Decision · Program_Chairs · 2023-09-22

**Decision:**

Accept (Poster)

**Comment:**

## Summary

This paper expands the original ProteinGym v0.1 benchmark from the Transception paper with over 100 new DMS datasets, clinical datasets, and greatly expanded evaluation existing pretrained, MSA-based, inverse folding, and hybrid models.

## Strengths
- [originality] The datasets and models considered represent a large gain over ProteinGym v0.1.
- [quality] The paper is very well-written and clear. The reviewers helped the authors to clarify many points for a broader audience.
- [quality] Releasing the MSAs and structures used to make predictions is essential for reproducibility and to make the benchmark accessible to other researchers.
- [significance] ProteinGym v1.0 provides a useful and unique resource for developing methods in the important and impactful fields of protein engineering and design.

## Weaknesses
- [quality] I would have liked to see more diversity in the pretrained models included. For example, including ProteinBERT (trained on GO annotations) or CARP (CNN instead of transformer) would have added some different dimensions instead of just different flavors of transformers.
- [originality] I suppose it's not the first attempt at a protein fitness benchmark, as several reviewers are very intent on reminding us.
- [significance] The DMS assays included here tend not to be good for capturing some aspects of proteins that are of interest to engineers, such as substrate/product selectivity.

## Comment on recommendation with respect to scores

This paper was somewhat difficult to evaluate because the reviewer scores ranged from 2 - 9. However, when reading more carefully, the primary concerns of the reviewer scoring 2 were double-dipping and a lack of novelty, which the authors addressed during the rebuttal period by clarifying the language, adding many DMS datasets, and pointing out the clinical datasets.